# Neural-Logic Human-Object Interaction Detection

**Liulei Li**[1], **Jianan Wei**[2], **Wenguan Wang**[2]*, **Yi Yang**[2]

[1]ReLER, AAII, University of Technology Sydney     [2]CCAI, Zhejiang University

https://github.com/weijianan1/LogicHOI

## Abstract

The interaction decoder utilized in prevalent Transformer-based HOI detectors typically accepts pre-composed `human-object` pairs as inputs. Though achieving remarkable performance, such paradigm lacks feasibility and cannot explore novel combinations over entities during decoding. We present LOGICHOI, a new HOI detector that leverages neural-logic reasoning and Transformer to infer feasible interactions between entities. Specifically, we modify the self-attention mechanism in vanilla Transformer, enabling it to reason over the ⟨`human`, `action`, `object`⟩ triplet and constitute novel interactions. Meanwhile, such reasoning process is guided by two crucial properties for understanding HOI: *affordances* (the potential actions an object can facilitate) and *proxemics* (the spatial relations between humans and objects). We formulate these two properties in *first-order* logic and ground them into continuous space to constrain the learning process of our approach, leading to improved performance and zero-shot generalization capabilities. We evaluate LOGICHOI on V-COCO and HICO-DET under both normal and zero-shot setups, achieving significant improvements over existing methods.

## 1 Introduction

The main purpose of human-object interaction (HOI) detection is to interpret the intricate relationships between human and other objects within a given scene[1]. Rather than traditional visual *perception* tasks that focus on the recognition of objects or individual actions, HOI detection places a greater emphasis on *reasoning* over entities[2], and can thus benefit a wide range of scene understanding tasks, including image synthesis[3], visual question answering[4,5], and caption generation[6,7], *etc*.

Current top-leading HOI detection methods typically adopt a Transformer[8]-based architecture, wherein the ultimate predictions are delivered by an interaction decoder. Such decoder takes consolidated embeddings of predetermined `human-object` pairs[9–20] as inputs, and it solely needs to infer the action or interaction categories of given entity pairs. Though achieving remarkable performance over CNN-based work, such paradigm suffers from several limitations: **first**, the principal focus of them is to optimize the samples with known concepts, ignoring a large number of feasible combinations that were never encountered within the training dataset, resulting in poor zero-shot generalization ability[21]. **Second**, the human-object pairs are usually proposed by a simple MLP layer[2,22–37] or simultaneously constructed when predicting humans and objects[10–17], without explicit modeling of the complex relationships among subjects, objects, and the ongoing interactions that happened between them. **Third**, existing methods lack the reasoning ability, caused by ignoring the key nature of HOI detection, *i.e.*, the interaction should be mined between entities, but not pre-given pairs.

In light of the above, we propose to solve HOI detection via the integration of Transformer and logic-induced HOI relation learning, resulting in LOGICHOI, which enjoys the advantage of robust distributed representation of entities as well as principled symbolic reasoning[38, 39]. Though

---

*Corresponding Author: Wenguan Wang.

37th Conference on Neural Information Processing Systems (NeurIPS 2023).

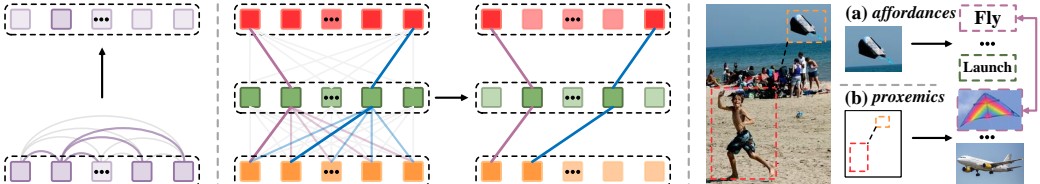

Figure 1: **Left**: self-attention aggregates information across pre-composed interaction (□) *queries*. **Middle**: in contrast, our proposed *triplet-reasoning* attention traverses over human (■), action (■), and object (■) *queries* to propose plausible interactions. **Right**: logic-induced *affordances* and *proxemics* knowledge learning.

it was claimed that Seq2Seq models (*e.g.*, Transformer) are inefficient to tackle visual *reasoning* problem[40, 41], models based on Transformer with improved architectures and training strategies have already been applied to tasks that require strong *reasoning* ability such as Sudoku[42], Raven's Progressive Matrices[43], compositional semantic parsing[44], spatial relation recognition[45], to name a few, and achieve remarkable improvements. Given the above success, we would like to argue that "Transformers are non-trivial symbolic reasoners, with only *slight architecture changes* and *appropriate learning guidances*", where the later imposes additional constraints for the learning and reasoning of modification. Specifically, for *architecture changes*, we modify the attention mechanism in interaction decoder that sequentially associates each counterpart of the current *query* individually (Fig. 1: left), but encourage it to operate in a triplet manner where states are updated by combining ⟨human, action, object⟩ three elements together, leading to *triplet-reasoning attention* (Fig. 1: middle). This allows our model to directly reason over entity and the interaction they potentially constituted, therefore enhancing the ability to capture the complex interplay between humans and objects. To *appropriately guide* the learning process of such triplet reasoning in Transformer, we explore two HOI relations, *affordances* and *proxemics*. The former refers to that an object facilitates only a partial number of interactions, and objects allowing for executing a particular action is predetermined. For instance, the human observation of a kite may give rise to imagined interactions such as launch or fly, while other actions such as repair, throw are not considered plausible within this context. In contrast, *proxemics* concerns the spatial relationships between humans and objects, *e.g.*, when a human is positioned below something, the candidate actions are restricted to airplane, kite, *etc*. This two kind of properties are stated in *first-order* logical formulae and serve as optimization objectives of the outputs of Transformer. After multiple layers of reasoning, the results are expected to adhere to the aforementioned semantics and spatial knowledge, thereby compelling the model to explore and learn the reciprocal relations between objects and actions, eventually producing more robust and logical sound predictions. The integration of logic-guided knowledge learning serves as a valuable complement to *triplet-reasoning attention*, as it constrains *triplet-reasoning attention* to focus on rule-satisfied triplets and discard unpractical combinations, enabling more effective and efficient learning, as well as faster convergence.

By enabling reasoning over entities in Transformer and explicitly accommodating the goal of promoting such ability through logic-induced learning, our method holds several appealing facets: **first**, accompanying reasoning-oriented architectural design, we embed *affordances* and *proxemics* knowledge into Transformer in a logic-induced manner. In this way, our approach is simultaneously a continuous neural computer and a discrete symbolic reasoner (albeit only implicitly), which meets the formulation of human cognition[46, 47]. **Second**, compared to neuro-symbolic methods solely driven by loss constraints[48–50], which are prone to be over-fitting on supervised learning[42] and disregard reasoning at inference stage, we supplement it with task-specific architecture changes to facilitate more flexible inference (*i.e.*, *triplet-reasoning* attention). Additionally, the constraints are tailored to guide the learning of interaction decoders rather than the entire model, to prevent "cheating". **Third**, our method does not rely on any discrete symbolic reasoners (*e.g.*, MLN[51] or ProbLog[52]) which increases the complexity of the model and cannot be jointly optimized with neural networks. **Fourth**, entities fed into Transformer are composed into interactions by the model automatically, such a compositional manner contributes to improved zero-shot generalization ability.

To the best of our knowledge, we are the first that leverages Transformer as the interaction *reasoner*. To comprehensively evaluate our method, we experiment it on two gold-standard HOI datasets (*i.e.*, V-COCO[53] and HICO-DET[54]), where we achieve **35.47%** and **65.0%** overall mAP score, setting new state-of-the-arts. We also study the performance under the zero-shot setup from four different perspectives. As expected, our algorithm consistently delivers remarkable improvements, up to **+5.16%** mAP under the *unseen object* setup, outperforming all competitors by a large margin.

## 2 Related Work

**HOI Detection.** Early CNN-based solutions for HOI detection can be broadly classified into two paradigms: two-stage and one-stage. The two-stage methods[2, 23–37, 55, 56] first detect entities by leveraging off-the-shelf detectors (*e.g.*, Faster R-CNN[57]) and then determine the dense relationships among all possible human-object pairs. Though effective, they suffer from expensive computation due to the sequential inference architecture[17], and are highly dependent on prior detection results. In contrast, the one-stage methods[58–61] jointly detect human-object pairs and classify the interactions in an end-to-end manner by associating humans and objects with predefined anchors, which can be union boxes[58, 61] or interaction points[59, 60]. Despite featuring fast inference, they heavily rely on hand-crafted post-processing to associate interactions with object detection results[10]. Inspired by the advance of DETR[62], recent work[9–20] typically adopts a Transformer-based architecture to predict interactions between humans and objects, which eliminates complicated post-processing and generally demonstrates better speed-accuracy trade-off. To learn more effective HOI representations, several studies[17, 19, 63–66] also seek to transfer knowledge from visual-linguistic pre-trained models (*e.g.*, CLIP[67]), which also enhances the zero-shot discovery capacity of models.

Although promising results have been achieved, they typically directly feed the pre-composed union representation of human-object pairs to the Transformer to get the final prediction, lacking *reasoning* over different entities to formulate novel combinations, therefore struggling with long-tailed distribution and adapting to *unseen* interactions. In our method, we handle HOI detection by *triplet-reasoning* within Transformer, where the inputs are entities. Such reasoning process is also guided by logic-induced *affordances* and *proxemics* properties learning with respect to the semantic and spatial rules, so as to discard unfeasible HOI combinations, and enhance zero-shot generalization abilities.

**Neural-Symbolic Computing** (NSC) is a burgeoning research area that seeks to seamlessly integrate the symbolic and statistical paradigms of artificial intelligence, while effectively inheriting the desirable characteristics of both[39]. Although the roots of NSC can be traced back to the seminal work of McCulloch and Pitts in 1943[68], it did not gain a systematic study until the 2000s, when a renewed interest in combining neural networks with symbolic reasoning emerged[51, 69–73]. These early methods often reveal limitations when attempting to handle large-scale and noisy data[74], as the effectiveness is impeded by highly hand-crafted rules and architectures. Recently, driven by both theoretical and practical perspectives, NSC has garnered increasing attention in holding the potential to model human cognition[38, 47, 75], combine modern numerical connectionist with logic reasoning over abstract knowledge [48, 76–81], so as to address the challenges of data efficiency [82, 83], explainability[84–87], and compositional generalization[88, 89] in purely data-driven AI systems.

In this work, we address neuro-symbolic computing from two perspectives, **first**, we employ Transformer as the symbolic *reasoner* but with slight architectural changes, *i.e.*, modifying the *self-attention* to *triplet-reasoning attention* and the inputs are changed from interaction pairs to `human`, `action`, and `object` entities, so as to empower the Transformer with strong *reasoning* ability for handling such relation interpretation task. **Second**, we condense *affordances* and *proxemics* properties into logical rules, which are further served to constrain the learning and reasoning of the aforementioned Transformer *reasoner*, making the optimization goal less ambiguous and knowledge-informed.

**Compositional Generalization**, which pertains to the ability to understand and generate a potentially boundless range of novel conceptual structures comprised of similar constituents[90], has long been thought to be the cornerstone of human intelligence[91]. For example, human can grasp the meaning of *dax twice* or *dax and sing* by learning the term *dax*[40], which allows for strong generalizations from limited data. In natural language processing, several efforts[92–100] have been made to endow neural networks with this kind of zero-shot generalization ability. Notably, the task proposed in [40], referred to as `SCAN`, involves translating commands presented in simplified natural language (*e.g.*, *dax twice*) into a sequence of navigation actions (*e.g.*, `I_DAX, I_DAX`). Active investigations into visual compositional learning also undergo in the fields of image caption[6, 7, 101, 102] and visual question answering[4, 5, 103, 104]. For instance, to effectively and explicitly ground entities, [102] first creates a template with slots for images and then fills them with objects proposed by open-set detectors.

Though there has already been work[21, 31, 32, 105–107] rethinking HOI detection from the perspective of compositional learning, they are, **i)** restricted to scenes with single object[105], **ii)** utilizing compositionality for data augmentation[31, 106] or representation learning[32], without generalization during learning nor relation reasoning. A key difference in our work is that our method

arbitrarily constitute interactions as final predictions during decoding. This compositional learning and inference manner benefit the generalization to *unseen* interactions. Moreover, the *affordances* and *proxemics* properties can be combined, *i.e.*, it is natural for us to infer that when a human is `above` a `horse`, the viable interactions options can only be `sit_on` and `ride`.

## 3   Methodology

In this section, we first review Transformer-based HOI detection methods and the self-attention mechanism (§3.1). Then, we elaborate the pipeline of our method and the proposed triplet-reasoning attention blocks (§3.2), which is guided by the logic-induced learning approach utilizing both *affordances* and *proxemics* properties (§3.3). Finally, we provide the implementation details (§3.4).

### 3.1   Preliminary: Transformer-based HOI Detection

Enlightened by the success of DETR[62], recent state-of-the-arts[9–20] for HOI detection typically adopt the encoder-decoder architecture based on Transformer. The key motivation shared across all of the above methods is that Transformer can effectively capture the long range dependencies and exploit the contextual relationships between human-object pairs[10, 12] by means of self-attention. Specifically, an interaction or action decoder is adopted, of which the *query*, *key*, *value* embeddings $\boldsymbol{F}^q, \boldsymbol{F}^k, \boldsymbol{F}^v \in \mathbb{R}^{N \times D}$ are constructed from the unified embedding of human-object pair $\boldsymbol{Q}^{h\text{-}o}$ by:

$$\boldsymbol{F}^q = (\boldsymbol{X} + \boldsymbol{Q}^{h\text{-}o}) \cdot \boldsymbol{W}^q, \quad \boldsymbol{F}^k = (\boldsymbol{X} + \boldsymbol{Q}^{h\text{-}o}) \cdot \boldsymbol{W}^k, \quad \boldsymbol{F}^v = (\boldsymbol{X} + \boldsymbol{Q}^{h\text{-}o}) \cdot \boldsymbol{W}^v, \tag{1}$$

where $\boldsymbol{X}$ is the input matrix, $\boldsymbol{W}^q, \boldsymbol{W}^k, \boldsymbol{W}^v \in \mathbb{R}^{D \times D}$ are parameter matrices and $\boldsymbol{Q}^{h\text{-}o}$ can be derived from the feature of union bounding box of human-object[22] or simply concatenating the embeddings of human and object together[17]. Then $\boldsymbol{X}$ is updated through a self-attention layer by:

$$\boldsymbol{X}'_i = \boldsymbol{W}^{v'} \cdot \textstyle\sum_{n=1}^{N} \operatorname{softmax}(\boldsymbol{F}^q_i \cdot \boldsymbol{F}^k_n / \sqrt{D}) \cdot \boldsymbol{F}^v_n. \tag{2}$$

Here we adopt the single-head variant for simplification. Note that under this scheme, the attention is imposed over action or interaction embeddings, which has already been formulated before being fed into the action or interaction decoder. This raises two concerns **i)** may discard positive human- object pair and **ii)** cannot present novel combinations over entities during decoding.

### 3.2   HOI Detection via Triplet-Reasoning Attention

In contrast to the above, we aim to facilitate the attention over three key elements to formulate an interaction (*i.e.*, `human`, `action`, `object`, therefore referring to *triplet-reasoning attention*), by leveraging the Transformer architecture. The feasible ⟨`human`, `action`, `object`⟩ tuples are combined and filtered through the layer-wise inference within the Transformer. Towards this goal, we first adopt a visual encoder which consists of a CNN backbone and a Transformer encoder $\mathcal{E}$ to extract visual features $\boldsymbol{V}$. Then, the learnable human *queries* $\boldsymbol{Q}^h \in \mathbb{R}^{N_h \times D}$, action *queries* $\boldsymbol{Q}^a \in \mathbb{R}^{N_a \times D}$, and object *queries* $\boldsymbol{Q}^o \in \mathbb{R}^{N_o \times D}$ are fed into three parallel Transformer decoders $\mathcal{D}^h, \mathcal{D}^a, \mathcal{D}^o$ to get the human, action, and object embeddings respectively by:

$$\boldsymbol{Q}^h = \mathcal{D}^h(\boldsymbol{V}, \boldsymbol{Q}^h), \quad \boldsymbol{Q}^a = \mathcal{D}^a(\boldsymbol{V}, \boldsymbol{Q}^a), \quad \boldsymbol{Q}^o = \mathcal{D}^o(\boldsymbol{V}, \boldsymbol{Q}^o). \tag{3}$$

Here, $N_h = N_a = N_o$ and the superscripts are kept for improved clarity. All of the three query embeddings are then processed by linear layers to get the final predictions. For human and object *queries*, we supervise it with the class and bounding box annotations, while for the action *queries*, we only supervise them with image-level categories, *i.e.*, what kinds of actions are happened in this image. After that, we adopt an interaction decoder $\mathcal{D}^p$ composed by multiple Transformer layers in which the self-attention is replace by our proposed *triplet-reasoning attention*, so as to empower Transformer with the *reasoning* ability. Specifically, in contrast to Eq. 1, given $\boldsymbol{Q}^h, \boldsymbol{Q}^a, \boldsymbol{Q}^o$, the input *query*, *key*, *value* embeddings $\boldsymbol{F}^q, \boldsymbol{F}^k, \boldsymbol{F}^v$ for *triplet-reasoning attention* are computed as:

$$\boldsymbol{F}^q = (\boldsymbol{X} + \boldsymbol{Q}^h + \boldsymbol{Q}^a) \cdot \boldsymbol{W}^q \in \mathbb{R}^{N_h \times N_a \times D},$$
$$\boldsymbol{F}^k = (\boldsymbol{X} + \boldsymbol{Q}^a + \boldsymbol{Q}^o) \cdot \boldsymbol{W}^k \in \mathbb{R}^{N_a \times N_o \times D}, \tag{4}$$
$$\boldsymbol{F}^v = \boldsymbol{W}^v_h \cdot (\boldsymbol{X} + \boldsymbol{Q}^h + \boldsymbol{Q}^a) \odot (\boldsymbol{X} + \boldsymbol{Q}^a + \boldsymbol{Q}^o) \cdot \boldsymbol{W}^v_o \in \mathbb{R}^{N_h \times N_a \times N_o \times D},$$

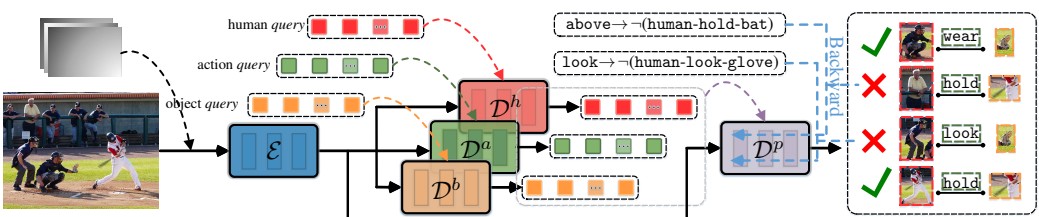

Figure 2: Overview of LOGICHOI. We first retrieve human, action, and object *queries* by $\mathcal{D}^h$, $\mathcal{D}^a$, and $\mathcal{D}^o$, respectively. Then $\mathcal{D}^p$ take them as input, reasoning over entities and combining potential interaction triplets. Finally, such process is guided by *affordances* and *proxemics* properties, to be more efficient and knowledge-informed.

where $\odot$ is the element-wise production. Note that we omit the dimension expanding operation for better visual presentation. Concretely, for $\boldsymbol{F}^q = (\boldsymbol{X}+\boldsymbol{Q}^h+\boldsymbol{Q}^a)\cdot\boldsymbol{W}^q$, the human *queries* $\boldsymbol{Q}^h \in \mathbb{R}^{N_h \times D}$ and action *queries* $\boldsymbol{Q}^a \in \mathbb{R}^{N_a \times D}$ are expanded to $\mathbb{R}^{N_h \times 1 \times D}$ and $\mathbb{R}^{1 \times N_a \times D}$, respectively. In this manner, $\boldsymbol{Q}^h+\boldsymbol{Q}^a$ associates each `human` and `action` entity, resulting in $N_h \times N_a$ `human-action` pairs in total. $N_a \times N_o$ viable `action-object` pairs are risen by $\boldsymbol{Q}^a+\boldsymbol{Q}^o$ in the same way. For the *value* embedding $\boldsymbol{F}^v$, it encodes the representation of all $N_h \times N_a \times N_o$ potential interactions. Given a specific element in $\boldsymbol{F}^v$, for instance, $\boldsymbol{F}^v_{inj}$, it is composed from $\boldsymbol{F}^q_{in}$ and $\boldsymbol{F}^k_{nj}$, corresponding to a feasible interaction of embeddings $\boldsymbol{Q}^h_i$, $\boldsymbol{Q}^a_n$, and $\boldsymbol{Q}^o_j$. Then each element in inputs $\boldsymbol{X}$ is updated by:

$$\boldsymbol{X}'_{ij} = \boldsymbol{W}^{v'}\cdot\sum_{n=1}^{N_a}\mathrm{softmax}(\boldsymbol{F}^q_{in}\cdot\boldsymbol{F}^k_{nj}/\sqrt{D})\cdot\boldsymbol{F}^v_{inj}, \qquad (5)$$

where $\boldsymbol{X}'$ denotes the output of *triplet-reasoning* attention. In contrast to *self-attention* (*cf*., Eq. 2), our proposed *triplet-reasoning attention* (*cf*., Eq. 5) stretches edges between every `human-action` and `action-object` pairs sharing the identical action *query*. By aggregating information from the relation between `human-action` and `action-object`, it learns to capture the feasibility of tuple $\langle$`human, action, object`$\rangle$ in a compositional learning manner, simultaneously facilitating reasoning over entities. The final output $\boldsymbol{Y}$ of $\mathcal{D}^p$ is given by:

$$\boldsymbol{Y} = \mathcal{D}^p(\boldsymbol{V}, \boldsymbol{Q}^h, \boldsymbol{Q}^a, \boldsymbol{Q}^o) \in \mathbb{R}^{N_h \times N_o \times D}, \qquad (6)$$

which delivers the interaction prediction for $N_h \times N_o$ `human-object` pairs in total. We set $N_h$, $N_a$, $N_o$ to a relatively small number (*e.g.*, 32) which is enough to capture the entities in a single image, and larger number of queries will exacerbate the imbalance between positive and negative samples. Additionally, for efficiency, we keep only half the number of human, object and action *queries* by filtering the low-scoring ones before sending them into the interaction decoder $\mathcal{D}^p$.

### 3.3 Logic-Guided Learning for Transformer-based Reasoner

In this section, we aim to guide the learning and reasoning of LOGICHOI with the *affordances* and *proxemics* properties. Though there has already been several works concerning about these two properties[15, 33, 63], they typically analyze *affordances* from the statistical perspective, *i.e.*, computing the distribution of co-occurrence of actions and objects so as to reformulate the predictions[33], simply integrate positional encodings into network features[15, 22], or proposing a two-path feature generator[63] which introduces additional parameters. In contrast, we implement them from the perspective that the constrained subset is the logical consequence of pre-given objects or actions.

Concretely, we first state these two kinds of properties as logical formulas, and then ground them into continuous space to instruct the learning and reasoning of our Transformer reasoner (*i.e.*, interaction decoder $\mathcal{D}^p$). For *proxemics*, we define five relative positional relationships with human as the reference frame, which are `above` (*e.g.*, kite `above` human), `below` (*e.g.*, skateboard `below` human), `around` (*e.g.*, giraffe `around` human), `within` (*e.g.*, handbag `within` human), `containing` (*e.g.*, bus `containing` human). To make the Transformer reasoner spatiality-aware, the human and object embeddings retrieved from Eq. 3 are concatenated with sinusoidal positional encodings generated from predicted bounding boxes. Then, given `action` $v$ and `position` relationship $p$, the set of infeasible interactions (*i.e.*, triplet $\langle$`human, action, object`$\rangle$) $\{h_1, \cdots, h_M\}$ can be derived:

$$\forall x(v(x) \wedge p(x) \rightarrow \neg h_1(x) \wedge \neg h_2(x) \wedge \cdots \wedge \neg h_M(x)), \qquad (7)$$

where $x$ refer to one `human-object` pair that is potential to have interactions. In first-order logic, the semantics of *variables* (*e.g.*, $x$) is usually referred to *predicates* (*e.g.*, `launch`$(x)$, `above`$(x)$). Eq. 7

states that, for instance, if the $v$ is `launch`, and $p$ is `above`, then in addition to interactions composed of actions apart from `launch`, `human-launch-boat` should be included in $\{h_1, \cdots, h_N\}$ as well. Similarly, given the `object` category $o$ and `position` relationship $p$, we shall have:

$$\forall x(o(x) \wedge p(x) \rightarrow \neg h_1(x) \wedge \neg h_2(x) \wedge \cdots \wedge \neg h_N(x)). \tag{8}$$

With Eq.7 and Eq.8, both *affordances* and *proemics* properties, and the combination relationship between them are clearly stated. Next we investigate how to convert the above logical symbols into differentiable operation. Specifically, logical connectives (e.g., $\rightarrow, \neg, \vee, \wedge$) defined on discrete Boolean variables are grounded to functions on continuous variables using product logic[108]:

$$
\begin{aligned}
\psi \rightarrow \phi = 1 - \psi + \psi \cdot \phi, &\qquad \neg \psi = 1 - \psi, \\
\psi \vee \phi = \psi + \phi - \psi \cdot \phi, &\qquad \psi \wedge \phi = \psi \cdot \phi.
\end{aligned}
\tag{9}
$$

Similarly, the *quantifier* are implemented in a generalized-mean manner following[109]:

$$
\begin{aligned}
\exists x(\psi(x)) &= \left(\frac{1}{K} \sum_{k=1}^{K} \psi(x_k)^q\right)^{\frac{1}{q}}, \\
\forall x(\psi(x)) &= 1 - \left(\frac{1}{K} \sum_{k=1}^{K} (1 - \psi(x_k))^q\right)^{\frac{1}{q}},
\end{aligned}
\tag{10}
$$

Given above relaxation, we are ready to translate properties defined in *first-order* logical formulae into sub-symbolic numerical representations, so as to supervise the interactions $\{h_1, \cdots, h_M\}$ predicted by the Transformer reasoner. For instance, Eq.7 is grounded by Eq.9 and Eq.10 into:

$$\mathcal{G}_{v,p} = 1 - \frac{1}{M} \sum_{m=1}^{M} \left(\frac{1}{K} \sum_{k=1}^{K} (s_k[v] \cdot s_k[h_m])\right), \tag{11}$$

where $s_k[v]$ and $s_k[h_m]$ refer to the scores of action $v$ and interaction $h_m$ with respect to input sample $x_k$. Here $K$ refers to the number of all training samples and we relax it to that in a mini-batch. As mentioned above, the spatial locations of human and objects are concatenated into the *query* which means the spatial relation is predetermined and can be effortlessly inferred from the box predictions (details provided in *Supplementary Materials*). Thus, we omit $p(x)$ in Eq.11. Then, the `action-position` loss is defined as: $\mathcal{L}_{v,p} = 1 - \mathcal{G}_{v,p}$. In a similar way, we can ground Eq.8 into:

$$\mathcal{G}_{o,p} = 1 - \frac{1}{N} \sum_{n=1}^{N} \left(\frac{1}{K} \sum_{k=1}^{K} (s_k[v] \cdot s_k[h_n])\right), \tag{12}$$

where $s_k[o]$ refers to the score of object regarding to the input sample $x_k$. The `object-position` loss is defined as: $\mathcal{L}_{o,p} = 1 - \mathcal{G}_{o,p}$. For $\mathcal{G}_{v,p}$, it scores the satisfaction of predictions to rules defined in Eq.7. For example, given a high probability of action `ride` (*i.e.*, a high value of $s_k[v]$) and the position relationship is `above`, if the probability of infeasible interactions (*e.g.*, `human-feed-fish"`) is also high, then $\mathcal{G}_{v,p}$ would receive a low value so as to punish this prediction. $\mathcal{G}_{o,p}$ is similar but it scores the satisfaction of predictions to Eq.8 with given position and objects such as `horse`, `fish`, *etc*. Through Eq.11 and Eq.12, we aim to achieve that, given the embeddings and locations of a group of human and object entities, along with the potential actions happened within the image, the Transformer reasoner should speculate which pair of human-object engaged in what kind of interaction, while the prediction should respect to the rules defined in Eq.7 and Eq.8.

### 3.4 Implementation Details

**Network Architecture.** To make fair comparison with existing Transformer-based work[9–20], we adopt ResNet-50 as the backbone. The visual encoder $\mathcal{E}$ is implemented as six Transformer encoder layers, while the three parallel human, object and action decoders $\mathcal{D}^h$, $\mathcal{D}^a$, $\mathcal{D}^o$ are all constructed as three Transformer decoder layers. For the interaction decoder $\mathcal{D}^p$, we instantiate it with three Transformer decoder layers as well, but replacing *self-attention* with our proposed *triplet-reasoning attention*. The number of human, object, action *queries* $N_h$, $N_o$, $N_a$ is set to 32 for efficiency, and the hidden sizes of all the modules are set to $D = 768$. Since the state-of-the-art work[17, 19, 63–66] usually leverages large-scale visual-linguistic pre-trained models to further enhance the detection capability, we follow this setup and adopt CLIP[67]. To improve the inference efficiency of our framework, we further follow [17] which uses guided embeddings to decode humans and objects in a single Transformer decoder, *i.e.*, merging $\mathcal{D}^h$ and $\mathcal{D}^o$ into a unified one to simultaneously output both human and object predictions.

**Training Objectives.** LOGICHOI is jointly optimized by the HOI detection loss (*i.e.*, $\mathcal{L}_{\text{HOI}}$) and logic-induced property learning loss (*i.e.*, $\mathcal{L}_{\text{LOG}}$):

$$\mathcal{L} = \mathcal{L}_{\text{HOI}} + \alpha\mathcal{L}_{\text{LOG}}, \quad \mathcal{L}_{\text{LOG}} = \mathcal{L}_{v,p} + \mathcal{L}_{o,p}. \tag{13}$$

Here $\alpha$ is set to 0.2 empirically. Note that $\mathcal{L}_{\text{LOG}}$ solely update the parameters of the Transformer reasoner (*i.e.*, interaction decoder $\mathcal{D}^p$) but not the entire network to prevent over-fitting. For $\mathcal{L}_{\text{HOI}}$, it is composed of human/object (*i.e.*, output of $\mathcal{D}^h$ and $\mathcal{D}^o$, respectively) detection loss, action (*i.e.*, output of $\mathcal{D}^a$) classification loss as well as interaction (*i.e.*, output of $\mathcal{D}^p$) classification loss.

## 4 Experiments

### 4.1 Experimental Setup

**Datasets.** We conduct experiments on two widely-used HOI detection benchmarks:
- V-COCO[53] is a carefully curated subset of MS-COCO[110] which contains 10,346 images (5,400 for training and 4,946 for testing). There are 263 human-object interactions annotated in this dataset in total, which are derived from 80 object categories and 29 action categories.
- HICO-DET[54] consists of 47,776 images in total, with 38,118 for training and 9,658 designated for testing. It has 80 object categories identical to those in V-COCO and 117 action categories, consequently encompassing a comprehensive collection of 600 unique human-object interactions.

**Evaluation Metric.** Following conventions[2, 10, 37], the mean Average Precision (mAP) is adopted for evaluation. Specifically, for V-COCO, we report the mAP scores under both scenario 1 (S1) which includes all of the 29 action categories and scenario 2 (S2) which excludes 4 human body motions without interaction to any objects. For HICO-DET, we perform evaluation across three category sets: all 600 HOI categories (Full), 138 HOI categories with less than 10 training instances (Rare), and the remaining 462 HOI categories (Non-Rare). Moreover, the mAP scores are calculated in two separate setups: **i)** the Default setup computing the mAP on all testing images, and **ii)** the Known Object setup measuring the AP for each object independently within the subset of images containing this object.

**Zero-Shot HOI Detection.** We follow the setup in previous work[17, 21, 31, 106, 107] to carry on zero-shot generalization experiments, resulting in four different settings: Rare First Unseen Combination (RF-UC), Non-rare First Unseen Combination (NF-UC), Unseen Verb (UV) and Unseen Object (UO) on HICO-DET. Specifically, the RF and NF strategies in the UC setting indicate selecting 120 most frequent and infrequent interaction categories for testing, respectively. In the UO setting, we choose 12 objects from 80 objects that are previously unseen in the training set, while in the UV setting, we exclude 20 verbs from a total of 117 verbs during training and only use them at testing.

**Training.** To ensure fair comparison with existing work[9–20], we initialize our model with weights of DETR[62] pre-trained on MS-COCO. Subsequently, we conducted training for 90 epochs using the Adam optimizer with a batch size of 16 and base learning rate $1e^{-4}$, on 4 GeForce RTX 3090 GPUs. The learning rate is scheduled following a "step" policy, decayed by a factor of 0.1 at the 60th epoch. In line with [10, 62, 66], the random scaling augmentation is adopted, *i.e.*, training images are resized to a maximum size of 1333 for the long edge, and minimum size of 400 for the short edge.

**Testing.** For fairness, we refrain from implementing

Table 1: Comparison of zero-shot generalization with state-of-the-arts on HICO-DET[54] `test`. See §4.2 for details.

| Method | VL Pretrain | Type | Unseen | Seen | Full |
|---|:---:|:---:|:---:|:---:|:---:|
| VCL [31][ECCV20] | | RF-UC | 10.06 | 24.28 | 21.43 |
| ATL [107][CVPR21] | | RF-UC | 9.18 | 24.67 | 21.57 |
| FCL [106][CVPR21] | | RF-UC | 13.16 | 24.23 | 22.01 |
| GEN-VLKT [17][CVPR22] | ✓ | RF-UC | 21.36 | 32.91 | 30.56 |
| SCL [21][ECCV22] | | RF-UC | 19.07 | 30.39 | 28.08 |
| LOGICHOI (ours) | ✓ | RF-UC | **25.97** | **34.93** | **33.17** |
| VCL [31][ECCV20] | | NF-UC | 16.22 | 18.52 | 18.06 |
| ATL [107][CVPR21] | | NF-UC | 18.25 | 18.78 | 18.67 |
| FCL [106][CVPR21] | | NF-UC | 18.66 | 19.55 | 19.37 |
| GEN-VLKT [17][CVPR22] | ✓ | NF-UC | 25.05 | 23.38 | 23.71 |
| SCL [21][ECCV22] | | NF-UC | 21.73 | 25.00 | 24.34 |
| LOGICHOI (ours) | ✓ | NF-UC | **26.84** | **27.86** | **27.95** |
| ATL [107][CVPR21] | | UO | 5.05 | 14.69 | 13.08 |
| FCL [106][CVPR21] | | UO | 0.00 | 13.71 | 11.43 |
| GEN-VLKT [17][CVPR22] | ✓ | UO | 10.51 | 28.92 | 25.63 |
| LOGICHOI (ours) | ✓ | UO | **15.67** | **30.42** | **28.23** |
| GEN-VLKT [17][CVPR22] | ✓ | UV | 20.96 | 30.23 | 28.74 |
| LOGICHOI (ours) | ✓ | UV | **24.57** | **31.88** | **30.77** |

Table 2: Quantitative results on HICO-DET[54] `test` and V-COCO[53] `test`. See §4.3 for details.

| Method | Backbone | VL Pretrain | Default | | | Known Object | | | $AP_{role}^{S1}$ | $AP_{role}^{S2}$ |
|---|---|---|---|---|---|---|---|---|---|---|
| | | | Full | Rare | Non-Rare | Full | Rare | Non-Rare | | |
| iCAN [111][BMVC18] | R50 | | 14.84 | 10.45 | 16.150 | 16.26 | 11.33 | 17.73 | 45.3 | - |
| UnionDet [58][ECCV20] | R50 | | 17.58 | 11.72 | 19.33 | 19.76 | 14.68 | 21.27 | 47.5 | 56.2 |
| PPDM [59][CVPR20] | HG104 | | 21.73 | 13.78 | 24.10 | 24.58 | 16.65 | 26.84 | - | - |
| HOTR [10][CVPR21] | R50 | | 23.46 | 16.21 | 25.60 | - | - | - | 55.2 | 64.4 |
| AS-Net [9][CVPR21] | R50 | | 28.87 | 24.25 | 30.25 | 31.74 | 27.07 | 33.14 | 53.9 | - |
| QPIC [11][CVPR21] | R50 | | 29.07 | 21.85 | 31.23 | 31.68 | 24.14 | 33.93 | 58.8 | 61.0 |
| CDN [13][NeurIPS21] | R50 | | 31.78 | 27.55 | 33.05 | 34.53 | 29.73 | 35.96 | 62.3 | 64.4 |
| MSTR [14][CVPR22] | R50 | | 31.17 | 25.31 | 32.92 | 34.02 | 28.83 | 35.57 | 62.0 | 65.2 |
| UPT [15][CVPR22] | R50 | | 31.66 | 25.94 | 33.36 | 35.05 | 29.27 | 36.77 | 59.0 | 64.5 |
| STIP [22][CVPR22] | R50 | | 32.22 | 28.15 | 33.43 | 35.29 | 31.43 | 36.45 | **65.1** | **69.7** |
| IF-HOI [18][CVPR22] | R50 | | 33.51 | 30.30 | 34.46 | 36.28 | 33.16 | 37.21 | 63.0 | 65.2 |
| ODM [112][ECCV22] | R50-FPN | | 31.65 | 24.95 | 33.65 | - | - | - | - | - |
| Iwin [113][ECCV22] | R50-FPN | | 32.03 | 27.62 | 34.14 | 35.17 | 28.79 | 35.91 | 60.5 | - |
| LOGICHOI (ours) | R50 | | **34.53** | **31.12** | **35.38** | **37.04** | **34.31** | **37.86** | 63.7 | 64.9 |
| CTAN [66][CVPR22] | R50 | ✓ | 31.71 | 24.82 | 33.77 | 33.96 | 26.37 | 36.23 | 60.1 | |
| SSRT [63][CVPR22] | R50 | ✓ | 30.36 | 25.42 | 31.83 | - | - | - | 63.7 | 65.9 |
| DOQ [65][CVPR22] | R50 | ✓ | 33.28 | 29.19 | 34.50 | - | - | - | 63.5 | - |
| GEN-VLK [17][CVPR22] | R50 | ✓ | 33.75 | 29.25 | 35.10 | 37.80 | 34.76 | 38.71 | 62.4 | 64.4 |
| LOGICHOI (ours) | R50 | ✓ | **35.47** | **32.03** | **36.22** | **38.21** | **35.29** | **39.03** | 64.4 | 65.6 |

any data augmentation during testing. Specifically, we first select $K$ interactions with the highest scores and further filter them by applying NMS to retrieve the final predictions. Following the convention [11, 16, 17, 20, 66], we set $K$ to 100.

## 4.2 Zero-Shot HOI Detection

The comparisons of our method against several top-leading zero-shot HOI detection models [17, 21, 31, 106, 107] on HICO-DET `test` are presented in Table 1. It can be seen that LOGICHOI outperforms all of the competitors by clear margins across four different zero-shot setups.

**Unseen Combination.** As seen, LOGICHOI provides a considerable performance gain against existing methods. In particular, it outperforms the top-leading GEN-VLKT [17] by **4.61%** and **1.79%** in terms of mAP on *unseen* categories for RF and NF selections, respectively. These numerical results substantiate our motivation of empowering Transformer with the *reasoning* ability and guide the learning in a logic-induced compositional manner, rather than solely taking Transformer as an interaction classifier.

**Unseen Object.** Our approach achieves dominant results under the UO setting, surpassing other competitors across all metrics. Notably, it yields *unseen* mAP **15.67%** and *overall* mAP **28.23%**, while the corresponding scores for the SOTA methods [17, 107] are 5.05%, 13.08% and 10.51%, 25.63%, presenting an improvement up to **5.16%** mAP for *unseen* categories. This reinforces our belief that empowering Transformer with logic-guided reasoning ability is imperative and indispensable.

**Unseen Verb.** Our method also demonstrates superior performance in the UV setup. Concretely, it surpasses GEN-VLKT [17] by **3.61%** mAP on *unseen* categories and achieves **30.77%** overall scores.

All of the above improvement on zero-shot generalization confirms the effectiveness of our proposed Transformer reasoner which learns in a compositional manner, and is informed by *affordances* and *proxemics* knowledge to address novel challenges that was never encountered before.

## 4.3 Regular HOI Detection

**HICO-DET.** In Table 2, we present the results of LOGICHOI and other top-performing models under the normal HOI detection setup. Notably, on HICO-DET[54] `test`, our solution demonstrates a significant improvement over previous state-of-the-art[17], with a substantial margin of **1.72%/ 2.78%/1.12%** mAP for Full, Rare, and Non-Rare categories, under the Default setup. Moreover, in terms of Known Object, our method attains exceptional mAP scores of **38.21%/35.29%/39.03%**.

Table 3: Analysis of essential components of LOGI-CHOI on HICO-DET [54]. See §4.4 for details.

| TRA | LRL | Full | Rare | Non-Rare |
|-----|-----|------|------|----------|
|  |  | 31.87 | 26.14 | 33.29 |
| ✓ |  | 34.32 | 30.67 | 35.19 |
|  | ✓ | 33.26 | 29.53 | 34.56 |
| ✓ | ✓ | **35.47**↑3.60 | **32.03**↑5.89 | **36.22**↑2.93 |

Table 4: Analysis of LRL under the zero-shot setup of *unseen* categories. See §4.4 for details.

| Setting | RF-UC | UO | UV |
|---------|-------|-----|-----|
| TRA | 24.01 | 13.26 | 23.14 |
| $+\mathcal{L}_{v,p}$ | 25.22 | 15.32 | 23.68 |
| $+\mathcal{L}_{o,p}$ | 25.34 | 13.91 | 24.29 |
| LOGICHOI | **25.97**↑1.96 | **15.67**↑2.41 | **24.57**↑1.43 |

Table 5: Analysis of number of decoder layer and *query* on HICO-DET [54] `test`. See §4.4 for details.

| # of layers ($L$) | Full | Rare | Non-Rare |
|-------------------|------|------|----------|
| 2 | 34.61 | 30.72 | 35.54 |
| **3** | **35.47** | **32.03** | **36.22** |
| 4 | 35.37 | 31.96 | 36.09 |
| 6 | 35.61 | 32.13 | 36.39 |

| # of queries ($N$) | Full | Rare | Non-Rare |
|--------------------|------|------|----------|
| 16 | 35.06 | 31.36 | 35.94 |
| **32** | **35.47** | **32.03** | **36.22** |
| 64 | 35.26 | 31.65 | 36.06 |
| 128 | 34.67 | 30.98 | 35.53 |

(a) number of the interaction decoder layer.  (b) number of the *queries*.

**V-COCO.** As indicated by the last two columns of Table 2, we also compare LOGICHOI with competitive models on V-COCO[53] `test`. Despite the relatively smaller number of images and HOI categories in this dataset, our method still yields promising results, showcasing its effectiveness. In particular, it achieves a mean mAP score of **65.0%** across two scenarios.

## 4.4 Diagnostic Experiment

For in-depth analysis, we perform a series of ablative studies on HICO-DET[54] `test`.

**Key Component Analysis.** We first examine the efficiency of essential designs of LOGICHOI, *i.e.*, *triplet-reasoning attention* (TRA) and logic-guided reasoner learning (LRL), which is summarized in Table 3. Three crucial conclusions can be drawn. First, our proposed *triplet-reasoning attention* leads to significant performance improvements against the baseline across all the metrics. Notably, TRA achieves **4.53%** mAP improvement on Rare congeries, demonstrating the ability of our Transformer Reasoner to reason over entities and generate more feasible predictions. Second, we also observe compelling gains from incorporating logic-guided reasoner learning into the baseline, even with basic self-attention, affirming its versatility. Third, our full model LOGICHOI achieves the satisfactory performance, confirming the complementarity and effectiveness of our designs.

**Logic-Guided Learning.** We guide the learning of Transformer reasoner with two logic-induced properties. Table 4 reports the related scores of *unseen* categories under three zero-shot setups. The contributions of $\mathcal{L}_{v,p}$ and $\mathcal{L}_{o,p}$ are approximately equal in the RF-UC setup, since during training, all actions and objects can be seen and utilized to guide reasoning. On the other hand, under the UO and UV setups, the improvements heavily rely on $\mathcal{L}_{v,p}$ and $\mathcal{L}_{o,p}$ respectively, while the other one brings minor enhancements. Finally, the combination of them leads to LOGICHOI, the new state-of-the-art.

**Number of Decoder Layer.** We further examine the effect of the number of Transformer decoder layer used in $\mathcal{D}^p$. As shown in Table 5a, LOGICHOI achieves similar performance when $L$ is larger than 2. For efficiency, we set $L = 3$ which is the smallest among existing work[9–20].

**Number of Query.** Next we probe the impact of the number of *query* for human, object and action in Table 5b. Note that the number of these three kind of queries is identical and we refer it to $N$. The best performance is obtained at $N = 32$ and more *queries* lead to inferior performance.

Table 6: Comparison of parameters and running efficiency.

| Method | Backbone | Params | FLOPs | FPS |
|--------|----------|--------|-------|-----|
| Two-stages Detectors: |  |  |  |  |
| iCAN [111][BMVC18] | R50 | 39.8 | - | 5.99 |
| DRG [30][ECCV20] | R50-FPN | 46.1 | - | 6.05 |
| SCG [36][ICCV21] | R50-FPN | 53.9 | - | 7.13 |
| STIP [22][CVPR22] | R50 | 50.4 | - | 6.78 |
| One-stages Detectors: |  |  |  |  |
| PPDM [59][CVPR20] | HG104 | 194.9 | - | 17.14 |
| HOTR [10][CVPR21] | R50 | 51.2 | 90.78 | 15.18 |
| HOITrans [12][CVPR21] | R50 | 41.4 | 87.69 | 18.29 |
| AS-Net [9][CVPR21] | R50 | 52.5 | 87.86 |  |
| QPIC [11][CVPR21] | R50 | 41.9 | 88.87 | 16.79 |
| CDN-S [13][NeurIPS21] | R50 | 42.1 | - | 15.54 |
| GEN-VLK$_s$ [17][CVPR22] | R50 | 42.8 | 86.74 | 18.69 |
| LOGICHOI (ours) | R50 | 49.8 | 89.65 | 16.84 |

**Runtime Analysis.** The computational complexity of our *triplet-reasoning attention* is squared compared to *self-attention*. Towards this, we make some specific designs: **i)** both the number of *queries* and Transformer decoder layers of our method are the smallest when compared to existing work[9–20], **ii)** as specified in §3.2, we filter the human, action, object *queries* and only keep half of them for efficiency, and **iii)** *triplet-reasoning attention* introduces few additional parameters. As summarized in Table 6, above facets make LOGICHOI even smaller in terms of Floating Point Operations (FLOPs) and faster in terms of inference compared to most existing work.

## 5 Conclusion

In this work, we propose LOGICHOI, the first Transformer-based neuro-logic reasoner for HOI detection. Unlike existing methods relying on predetermined human-object pairs, LOGICHOI enables the exploration of novel combinations of entities during decoding, improving effectiveness as well as the zero-shot generalization capabilities. This is achieved by **i)** modifying the *self-attention* mechanism in vanilla Transformer to reason over ⟨human, action, object⟩ triplets, and **ii)** incorporating *affordances* and *proxemics* properties as logic-induced constraints to guide the learning and reasoning of LOGICHOI. Experimental results on two gold-standard HOI datasets demonstrates the superiority against existing methods. Our work opens a new avenue for HOI detection from the perspective of empowering Transformer with symbolic reasoning ability, and we wish it to pave the way for future research.

**Acknowledgement.** This work was partially supported by the Fundamental Research Funds for the Central Universities (No. 226-2022-00051) and CCF-Tencent Open Fund.

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
