# Neural-Logic Human-Object Interaction Detection
## *Supplementary Materials*

**Liulei Li**[1], **Jianan Wei**[2], **Wenguan Wang**[2]*, **Yi Yang**[2]

[1]ReLER, AAII, University of Technology Sydney    [2]CCAI, Zhejiang University

https://github.com/weijianan1/LogicHOI

This document provides additional materials to supplement our main manuscript. We first summarize extra implementation details of LOGICHOI in §A. Qualitative results as well as analysis on typical failure cases are provided in §B. Finally, we offer further discussion on the limitation and social impact of LOGICHOI in §C.

## A    More Implementation Detail

The detection loss used for the output of human decoder (*i.e.*, $\mathcal{D}^h$) and object decoder (*i.e.*, $\mathcal{D}^o$) is implemented in accordance with DETR[1]. Specifically, we compute the object classification loss, and adopt the $\ell_1$ loss as well as the generalized intersection over union (GIoU) loss for bounding box regression during training. The final prediction of interaction decoder (*i.e.*, $\mathcal{D}^p$) is the category of human-object interaction (*i.e.*, $\langle$human, action, object$\rangle$ triplet) rather than a single action since the inputs are three elements to construct the interaction and we aim to not only interpret the complex relation between them, but also refine the object and action predictions. To facilitate the visual knowledge transfer from CLIP[2], we follow previous work[3–8] to adopt the ViT-B/32 variant and freeze its weights during training. Moreover, an auxiliary loss is applied to the intermediate outputs of each decoder layer which contributes to improved results in the decoding process.

## B    Qualitative HOI Detection Result

We provide qualitative results of our method, including both success and failure cases in Fig. S2. It can be observed that our method demonstrates remarkable improvements in HOI detection across a wide range of scenarios. The integration of triplet reasoning and logic-guided knowledge learning enables our model to effectively capture intricate relationships between humans and objects, leading to enhanced detection accuracy. Nonetheless, there are certain scenarios where our method encounters challenges. Specifically, in the last column of Figure S2, we observe that our model faces difficulties when dealing with highly ambiguous relations, such as instances where a frisbee is held by a human in a strange pose. The complex spatial arrangement and occlusion make it challenging for the model to accurately infer the correct HOI. Additionally, our model may be inefficient when it needs to deduce additional contextual cues. For example, in cases where a chair is partially occluded by a human, the model may struggle to correctly recognize the interaction between the two entities due to the lack of complete visual information.

## C    Discussion

### C.1    Limitation

It is important to acknowledge a limitation regarding the scale of validation within our study. The number of interactions included in the dataset for model evaluation is limited to fewer than 600

---

*Corresponding Author: Wenguan Wang.

37th Conference on Neural Information Processing Systems (NeurIPS 2023).

Table S1: Comparison of efficiency and performance on HICO-DET[9] `test` and V-COCO[10] `test`.

| Method | Backbone | Params | FLOPs | FPS | Default Full | Default Rare | Default Non-Rare | $AP_{role}^{S1}$ | $AP_{role}^{S2}$ |
|---|---|---|---|---|---|---|---|---|---|
| Two-stages Detectors: | | | | | | | | | |
| iCAN [11][BMVC18] | R50 | 39.8 | - | 5.99 | 14.84 | 10.45 | 16.15 | 45.3 | - |
| DRG [12][ECCV20] | R50-FPN | 46.1 | - | 6.05 | 19.26 | 17.74 | 19.71 | 51.0 | - |
| SCG [13][ICCV21] | R50-FPN | 53.9 | - | 7.13 | 31.33 | 24.72 | 33.31 | 54.2 | 60.9 |
| STIP [14][CVPR22] | R50 | 50.4 | - | 6.78 | 32.22 | 28.15 | 33.43 | **65.1** | **69.7** |
| One-stages Detectors: | | | | | | | | | |
| PPDM [15][CVPR20] | HG104 | 194.9 | - | 17.14 | 21.73 | 13.78 | 24.10 | - | - |
| HOTR [16][CVPR21] | R50 | 51.2 | 90.78 | 15.18 | 25.10 | 17.34 | 27.42 | 55.2 | 64.4 |
| HOITrans [17][CVPR21] | R50 | 41.4 | 87.69 | 18.29 | 23.46 | 16.91 | 25.41 | 52.9 | - |
| AS-Net [18][CVPR21] | R50 | 52.5 | 87.86 | 17.21 | 28.87 | 24.25 | 33.14 | 53.9 | - |
| QPIC [19][CVPR21] | R50 | 41.9 | 88.87 | 16.79 | 29.07 | 21.85 | 31.23 | 58.8 | 61.0 |
| CDN-S [20][NeurIPS21] | R50 | 42.1 | - | 15.54 | 31.78 | 27.55 | 33.05 | 62.3 | 64.4 |
| GEN-VLK$_s$ [8][CVPR22] | R50 | 42.8 | 86.74 | 18.69 | 33.75 | 29.25 | 35.10 | 62.4 | 64.4 |
| LOGICHOI (ours) | R50 | 49.8 | 89.65 | 16.84 | **35.47** | **32.03** | **36.22** | 64.4 | 65.6 |

instances. This constrained sample size falls short of capturing the full spectrum of interactions that take place in real-world scenarios. Consequently, the exploration of applications related to object and interaction detection in more complex and diverse situations may be hindered.

## C.2  Broader Impact

This work provides a feasible way to interpret complex relationships between human beings and objects, and can thus benefit a variety of applications, including but not limited to robotics, health care, and autonomous driving, *etc*. Nevertheless, there is a risk that LOGICHOI would be used inappropriately, for instance, the constant monitoring and detection of human-object interactions may raise concerns about intrusive surveillance and the collection of personal data without consent. Therefore, it is imperative to duly consider ethical requirements and legal compliance when addressing the apprehensions regarding individual privacy. Meanwhile, in order to prevent potential negative social effects, it is crucial to develop robust security protocols and systems that effectively safeguard sensitive information, eliminating the risk of cyber attacks and data breaches.

## D  License

The V-COCO[10] and HICO-DET[9] datasets are released under the MIT license and the CC0: Public Domain license, respectively. We employ them for the purpose of research.

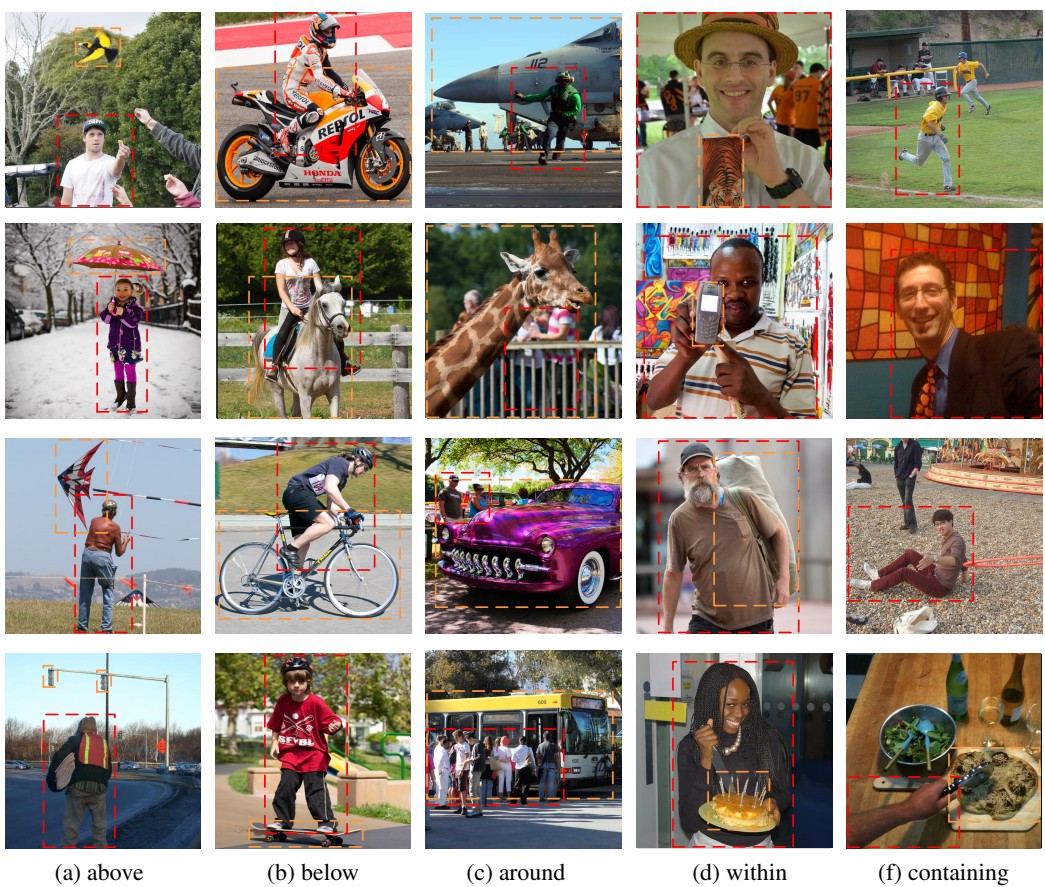

(a) above      (b) below      (c) around      (d) within      (f) containing

Figure S1: Examples of the five spatial relations from V-COCO[10] and HICO-DET[9].

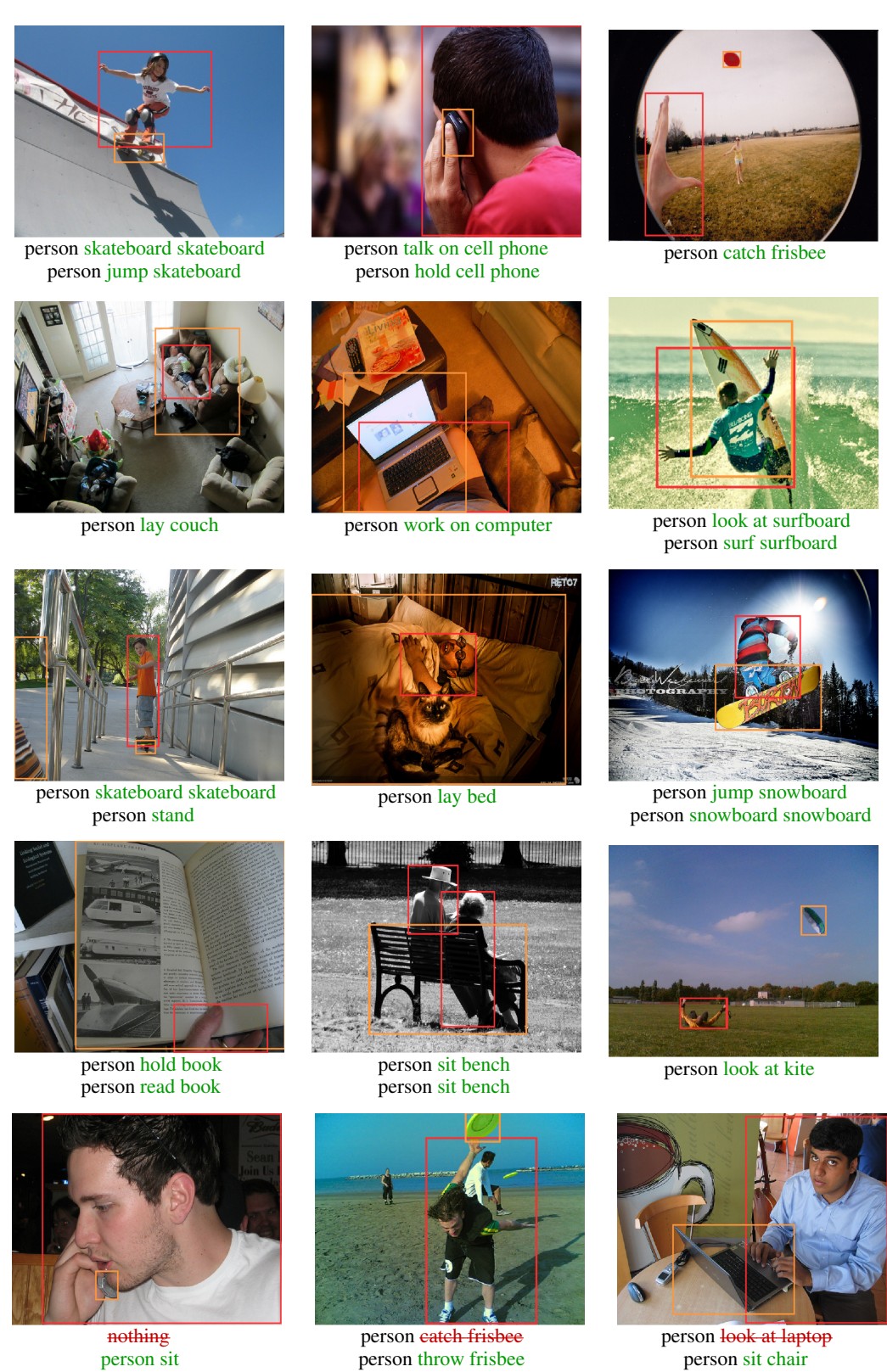

Figure S2: Successful and failure cases selected from V-COCO[10] and HICO-DET[9].