# OpenReview forum: "Neural-Logic Human-Object Interaction Detection"
_NeurIPS.cc/2023/Conference — NeurIPS 2023 poster_

### Official Review · Reviewer_uwRp · 2023-07-03

**Soundness:** 2 fair
**Presentation:** 3 good
**Contribution:** 2 fair
**Rating:** 4
**Confidence:** 4

**Summary:**

This paper introduces a novel Human-Object Interaction (HOI) detection method called Logic-Guided Transformer Reasoner (LoTR). Unlike standard Transformer-based HOI detectors that accept pre-composed human-object pairs, LoTR leverages Transformers to infer plausible interactions among entities, thus enabling the creation of novel interactions. This is achieved by modifying the self-attention mechanism to reason over human-action-object triplets. The reasoning process is guided by two critical elements of HOI: affordances and proxemics, expressed in first-order logic and grounded into a continuous space. These adaptations enhance the learning of LoTR, leading to improved performance and zero-shot generalization capabilities. Evaluations on V-COCO and HICO-DET datasets demonstrate LoTR's superiority over existing methods under normal and zero-shot conditions.

**Strengths:**

+ A novel perspective for addressing problems in the Human-Object Interaction (HOI) domain, which is inspiring.

+ An abundance of experimental evidence demonstrates that the method has yielded impressive results.

+ The logic-guided learning method in section 3.3 of the paper is interesting and can provide assistance for the development of other computer vision tasks.


**Weaknesses:**

-  Although the method has great advantages and is thought-provoking in the HOI (Human-Object Interaction) field, its application seems somewhat limited. Can it be applied to other visual tasks? If so, I am really looking forward to such interesting things happening.

**Questions:**

Please refer to the weaknesses.

**Limitations:**

I do not find any negative societal impact in this work.

---

> ### Author Rebuttal · Authors · 2023-08-09
>
> **Q: Applicability to visual tasks beyond HOI detection.**
>
> **A:** Thank you for your interest in the generability of LoTR. We firmly believe that the core principles of our logic-guided reasoning framework have the potential to be extended and applied to various computer vision tasks. This adaptability stems from:
>
>  **First.** Our approach facilitates entity-centric reasoning so as to make use of the relationships between them. This contributes to holistic scene understanding [ref1] and a series of work in different disciplines can benefit from it.
> - For example, the consideration of relationships among objects of distinct semantic classes holds the potential for refining **image classification** and **object detection**. For instance, the co-occurrence of items like a mouse and laptop within a single image [ref2, ref3] can be harnessed.
> - Moreover, scenes and objects can be deconstructed into compositional elements (*e.g.*, the living room consists of several objects like a sofa, coffee table, and sofa made from cushions, armrests, *etc*.)[ref4]. A nuanced grasp of these mereological relationships between multiple parts shall significantly bolster **scene/object parsing** performance [ref5].
> - Additionally, in high-level scene understanding tasks such as **image caption** or **visual question answering (VQA)**, to solve questions like "how many people are there near the bus?" the model should not only identify "people" and "car", but also have to infer the relationships between them, reasoning whether the person is near the car, or far away from it [ref6].
>
>  **Second.** We state world knowledge in first-order logic formulae and subsequently ground them into constraints towards guiding the model to generate more robust and logical sound predictions. This process serves as a vital complement to the entity-centric reasoning above. Since either the co-occurrence relation between objects or the topology of a specific scene/object is hard to directly retrieve by gradient optimization of training samples. Not to mention the extensive spectrum of human knowledge covering spatial arrangements, material properties, and even mathematical principles that are necessary for VQA. All of the above can be ground into constraints by our method and then assist the reasoning between different entities, *i.e.*, logic-guided reasoner learning.
>
> ---
>
> To demonstrate the discussion above, we experiment with our logic-guided reasoning framework on two tasks, *i.e.*, **weakly-supervised semantic segmentation** and **VQA**.
>
>
> For **weakly-supervised semantic segmentation (WSSS)**, it has been proved that reasoning over different objects belonging to the same category can make up for the absence of detailed supervision by mining the co-patterns between them [ref7, ref8]. Here we follow [ref5] which takes three images as a group and then employs the graph neural network to conduct recursive reasoning for semantic understanding. Specifically, we replace GNN in [ref8] with a Transformer decoder where the self-attention is replaced with our triplet-reasoning attention (denoted as TRA). Moreover, we constrain the mereological relationships of objects in the COCO dataset. For example, "knife", "fork", "spoon" usually appear in the kitchen, and "snowboard", "sports ball", "baseball bat" are all sports equipment (denoted as LRL). The results on COCO are summarized below:
>
>
> |Method|mIoU|
> |:----------|:----------|
> |Baseline [ref8]|24.3|
> |+ TRA|26.9|
> |+ LRL|27.8|
>
> Due to the limited time, the performance reported above is the mIoU of the pseudo labels. In WSSS, these pseudo labels will next serve as the ground truth to train a segmentation network that is the same for all methods (*e.g.*, DeepLabV3). Thus the performance of pseudo labels can already reveal the effectiveness of the proposed method to some extent.
>
>
> In the context of **VQA**, there have been continuous efforts towards leveraging knowledge to assist question answering by querying a pre-establish knowledge base [ref9]. Beyond the basic query operation, we further transform the knowledge into first-order logic formulae and use them to constrain the learning of models by our proposed logic-guided learning approach (denoted as LRL).
> The results on the OK-VQA dataset are summarized below:
>
> |Method|accuracy v1.0|
> |:----------|:----------|
> |Baseline [ref9]|38.35|
> |+ LRL|39.42|
>
> For fast implementation, we temporarily discard our triplet-reasoning attention here.
>
>
> Given the above experiments on two different tasks, though the baselines we used may not be state-of-the-arts, the solid gain in performance can still demonstrate the general effectiveness of our method and its promising capacity to impart a positive impact on the general computer vision community.
>
>
> [ref1] Adopting Abstract Images for Semantic Scene Understanding. TPAMI 2014
>
> [ref2] Structure Inference Net: Object Detection Using Scene-Level Context and Instance-Level Relationships. CVPR 2018.
>
> [ref3] Joint binary classifier learning for ECOC-based multi-class classification. TPAMI 2015.
>
> [ref4] Vision GNN: An Image is Worth Graph of Nodes. NeurIPS 2022.
>
> [ref5] Towards Total Scene Understanding: Classification, Annotation, and Segmentation in an Automatic Framework. CVPR 2009.
>
> [ref6] Visual Query Answering by Entity-Attribute Graph Matching and Reasoning. CVPR2019.
>
> [ref7] Mining cross-image semantics for weakly supervised semantic segmentation. ECCV 2020.
>
> [ref8] Group-Wise Semantic Mining for Weakly Supervised Semantic Segmentation. AAAI 2021.
>
> [ref9] KRISP: Integrating Implicit and Symbolic Knowledge for Open-Domain Knowledge-Based VQA. CVPR 2021.
>
> ---
>
> Thanks again for your interest in the broader applications of our method. Hope our response has addressed your question and further suggestions are welcome.

---

> > ### Author Response · Authors · 2023-08-20
> > **Looking forward to your further discussion**
> >
> > Dear Reviewer uwRp,
> >
> > Thank you again for your kind review and comments. We hope we have addressed all of your concerns. We wonder if there are any additional comments. We sincerely hope that we will be able to use the remaining time to engage in an open dialogue with domain experts to enhance the quality of our work.
> >
> > Thanks again, Authors

---

> > ### Comment · Area_Chair_Cwny · 2023-08-20
> > **Response Needed**
> >
> > Dear uwRp,
> >
> > Given you are in the minority with a reject it is critical that you read the rebuttal and inquire about any outstanding issues you feel are still not addressed.
> >
> > Thank you,
> > AC

---

### Official Review · Reviewer_gPtK · 2023-07-04

**Soundness:** 4 excellent
**Presentation:** 3 good
**Contribution:** 3 good
**Rating:** 6
**Confidence:** 4

**Summary:**

The paper introduces a novel approach to tackle the problem of human-object interaction detection. It presents two main contributions. Firstly, there is a revisited self-attention mechanism within the decoder that focuses on the interaction queries. This aspect has received limited attention in the field. Secondly, the paper introduces a logic-based reasoner that employs affordances and proxemics to filter out infeasible interactions. This reasoner replaces conventional hard-coded rule-based logic with differentiable operations. In my opinion, these technical advancements are original and are likely to be of interest to researchers in this field. Furthermore, experimental results on HICO-DET and V-COCO datasets demonstrate state-of-the-art performance.

**Strengths:**

- Most Transformer-based HOI detection methods rely on the vanilla self-attention mechanism borrowed from DETR [59]. DIfferently, this paper introduces a reasonable enhancement to better adapt for the combination nature of human-object interactions. The idea is original and likely to be of interest to researchers in this field.

- Some prior methods (e.g., [15][49]) have utilized rule-based logic to filter out infeasible human-object interactions, resulting in improved performance. This paper takes a further step by converting the rule-based logic into differentiable operations, which is very interesting.

- The paper is generally well-written. Readers with a relevant background will have no difficulty following the paper.

- The proposed method demonstrates significant improvements over existing approaches on both general and zero-shot settings.

**Weaknesses:**

There are a few minor weaknesses in the experimental section that should be addressed:

- It remain unclear whether the proposed Transformer reasoner can be applied to common HOI detectors and contribute to performance improvements across different methods, or if it is specifically designed for the approach in this paper. To provide a comprehensive understanding, it would be valuable for the paper to include additional results such as GEN-VLK+reasoner, QPIC+reasoner, CDN+reasoner, and others, to showcase the potential benefits of integrating the reasoner into other existing methods.

- In Section 4.2, there is a need for more in-depth analysis explaining why the proposed method can achieve better results on unseen objects and unseen verbs. Although an analysis on the reasoner is presented in L344-349, it remains unclear why TRA without the reasoner can also achieve state-of-the-art performance on the zero-shot setting, as shown in Table 4.

- Regarding the baseline results in Table 3, it appears that the baseline method has already achieved better performance compared to many existing methods. The baseline does not seem to be a vanilla model with the standard Transformer encoder-decoder architecture. It would be helpful to address these concerns by providing additional implementation details.

Typos:
- L333: examine the efficiency of … -> examine the effect of …?

- L45, L47: Fig.2 -> Fig.1

- Eq.(7): $\neg h_M(x)) \rightarrow \neg h_N(x)$

**Questions:**

Please refer to the above section for further details. It would be greatly appreciated if the following questions could be addressed:

- What are the potential benefits of integrating the reasoner into existing Transformer-based methods?

- A more in-depth analysis explaining why the proposed method achieves improved results on unseen objects and unseen verbs.

- Implementation details regarding the baseline method (as indicated in the first row of Table 3)?

**Limitations:**

The limitations have been discussed in the supplementary materials. The primary limitation arises from the computational complexity of the proposed method.

---

> ### Author Rebuttal · Authors · 2023-08-09
>
> **Q1: Generalize to more existing Transformer-based HOI detectors.**
>
> **A1:** Sorry for this confusion. Our approach involves three distinct elements as input to the interaction decoder $D^p$: human, object, and action. In this context, the action item globally encompasses all possible actions occurring within the image, but unlike existing work that one single action is derived from a specific human-object pair. Therefore, the reasoner here cannot be directly adapted to existing work, but should add an additional decoding branch to predicate all potential actions. However, to properly address your concern, we give experiments on two approaches ( *i.e.*, QPIC[11] and GEN-VLKT[17]) by introducing an extra action decoder as well as our reasoner to them. The results are given below.
>
> | Method    | Backbone| Full    | Rare    | Non-Rare|
> |:----------|:--------|:--------|:--------|:--------|
> | QPIC[11]  | R50     | 29.07   |21.85    |31.23    |
> | QPIC + reasoner |R50| **32.23** (+3.16)   |**25.97** (+4.12) |**33.81** (+2.58) |
> | GEN-VLKT[17]|R50    | 33.75   |29.25    |35.10    |
> | GEN-VLKT + reasoner | R50 | **35.13** (+1.38) | **31.50** (+2.25) |**36.13** (+1.03) |
>
> We do not heavily tune the hyper-parameters due to limited time, but there is still considerable improvement over these two methods which demonstrates the general effectiveness of our proposed logic-guided reasoner.
>
> ---
>
> **Q2: Analysis on improvement under the zero-shot setup.**
>
> **A2:** For the improvement on zero-shot setups brought by Triplet-reasoning Attention (TRA), we provide discussions in **L76-77** and **L138-140** from the perspective of compositional generalization. Specifically, in our method, we break HOI detection into two parts: i) predicting all humans and objects and actions present in the image, and ii) employing TRA to reason over these three kinds of entities (**i.e.**, "human", "object" and "action") and compose the possible interactions (**i.e.**, "human-action-object"). Such a process entails a combination phase over three elements to deliver the ultimate prediction, but not directly fit the annotations. This improves the generalizability of the model and contributes to enhanced zero-shot HOI detection ability.
>
> For the improvement brought by logic-guided reasoner learning (LRL), the affordances and proxemics properties are defined in an exclusive manner (i.e., the interactions can **NOT** be which categories with pre-given verb/object and position) in Eq.7 and Eq.8. As a result, even if the model has never encountered interactions like "human-launch-boat", the high possibility of action prediction "launch" excludes interaction predictions such as "human-wash-boat" and "human-drive-boat", eventually reinforcing the validity of the outcomes.
>
> To render a more intuitive understanding of the improvement brought by each component, Table 4 is updated to:
>
> | TRA     |$\mathcal{L}_{o,p}$|$\mathcal{L}_{v,p}$| RF-UC | NF-UC |UO     | UV      |
> |:--------|:------------------|:------------------|:------|:------|:------|:--------|
> |         |                   |                   | 20.63 | 24.35 |9.64   | 20.23   |
> | &#10003;|                   |                   | 24.01 | 25.72 |13.26  | 23.14   |
> |         |      &#10003;     |                   | 25.22 | 26.45 |15.32  | 23.68   |
> |         |                   | &#10003;          | 25.34 | 26.57 |13.91  | 24.29   |
> | &#10003;|      &#10003;     | &#10003;          | 25.97 | 26.84 |15.67  | 24.57   |
>
> As seen, TRA can boost the baseline methods up to **3.38** on RF-UC, **1.37** on NF-UC, **3.62** on UO, and **2.91** on UV, respectively. Such a solid improvement supports our motivation to address HOI detection from the perspective of compositional generalization.
>
> Table 4 and related parts will be updated.
>
> ---
>
> **Q3: Implementation details of the baseline method.**
>
> **A3:** As specified in **A1** that the input to interaction decoder $D^p$ of LoTR contrasts significantly to the pipeline of existing work, we do not rely on the implementation of a specific approach but refer to a series of existing work. For instance, the architectural blueprints of human/object and interaction decoders adhere closely to CDN-S[13] which also merely adopts three Transformer decoding layers; the implementation of the action branch draws inspiration from [16]; the ultimate prediction follows GEN-VLKT[17] which delivers the class of interaction (*i.e.*, "human-action-object") but not a single action. The performance of the resultant baseline method is given below.
>
> | Method  | Backbone | Full | Rare | Non-Rare| AP$_\{role\}^\{S1\}$ | AP$_\{role\}^\{S2\}$
> |:--------|:---------|:-----|:-----|:--------|:--------|:--------|
> |baseline | R50      | 31.87| 26.14| 33.29   | 62.1    | 63.6    |
> |LoTR     | R50      | 35.47(**+3.70**)| 32.03(**+5.89**)| 36.22(**+2.93**)   | 64.4(**+2.3**)   | 65.6(**+2.0**)    |
>
> As seen, despite the baseline having relatively higher performance compared to certain existing methods, LoTR still exhibits substantial improvements over the baseline (*i.e.*, **+3.7** on HICO-DET and **+2.15** on V-COCO), which is remarkable.
>
> To ensure reproducibility, the implementation details will be enriched in *Supplementary Material* and our code shall be released upon acceptance.
>
> ---
>
> **Q4: Typo.**
>
> **A4:** Sorry for these mistakes. We will revise them in the new version.
>
> ---
>
>
> We greatly appreciate you for your valuable time and constructive feedback. Hope our response has addressed your concerns and if you have any further questions, please feel free to raise them. We will do our utmost to give the clarification.

---

> > ### Author Response · Authors · 2023-08-20
> > **Looking forward to your further discussion**
> >
> > Dear Reviewer gPtK,
> >
> > Thank you again for your kind review and comments. We hope we have addressed all of your concerns. We wonder if there are any additional comments. We sincerely hope that we will be able to use the remaining time to engage in an open dialogue with domain experts to enhance the quality of our work.
> >
> > Thanks again, Authors

---

### Official Review · Reviewer_jaUK · 2023-07-05

**Soundness:** 3 good
**Presentation:** 3 good
**Contribution:** 3 good
**Rating:** 5
**Confidence:** 5

**Summary:**

This paper introduces LoTR: Logic-Guided Transformer Reasoner for Human-Object Interaction (HOI) detection that adapts the self-attention mechanism in standard Transformers to infer feasible interactions over entities, guided by affordances and proxemics.

**Strengths:**

none

**Weaknesses:**

1-The meanings of the notations in the paper are confusing. For instance, in Equation 2, the meaning of $W^{v\'}$ is unclear, and there is no clarification provided in the text. It would be helpful if the authors could elaborate on the meaning of $X_i$. Since Equation 2 forms a significant contrast to the proposed triplet-reasoning attention (Equation 5), it should be thoroughly explained for improved readability.

2-In Equation 5, the authors derive $X_{ij}$. It would be beneficial if they could provide a detailed explanation of how $X_{ij}$ differs from $X_i$.

3-In Line 213, $h_N$ should be $h_M$. Furthermore, Equations 7 and 8 both have an extra right parenthesis.

4-The method of designing a loss function using logical reasoning is interesting, but it seems rather complex. Could the authors use more intuitive language to describe $G_{v,p}$ and $G_{o,p}$?

5-The introduction of logic-induced affordances and proxemics, in essence, imposes additional constraints on spatial positions within HOI. Could this lead to overfitting on these two datasets, thereby reducing the model's generalizability on a broader range of data? For example, I don't fully agree with the statement in Line 238, "if there is a low possibility of a horse and the human box is above the object box, then the reasoning results should not be 'human-feed-horse.'" This scenario is not entirely implausible as a person could feasibly feed a horse from atop it.

**Questions:**

How to obtain the set of infeasible interactions (i.e., triplet ⟨human, action, object⟩) {h1, · · · , h_N} and {h1, · · · , h_M}?

---

> ### Author Rebuttal · Authors · 2023-08-09
>
> **Q1: Clarify $\textbf{W}^{v'}$ and $\textbf{X}_i$ in Eq.2.**
>
> **A1:** Sorry for this confusion. Here $\textbf{W}^{v'}$ is the weight of a linear layer that adapts the outputs of the attention mechanism. For $\textbf{X}_i$ in Eq.2, it refers to the soft weight assigned to each **pre-constituted** human-object pair $i$, as the query $\textbf{F}_i^q$ and key $\textbf{F}_n^k$ are mapped from unified embeddings of human-object pairs (*i.e.*, $\textbf{Q}^\{h-o\}$). For prior two-stage approaches, the **pre-constituted** human-object pairs are proposed by an MLP layer [21,39-53], and for one-stage methods, they are usually simultaneously constructed when predicting humans and objects [10-17].
>
> ---
>
> **Q2: Clarify how $\textbf{X}_{ij}$ in Eq.5 is different to $\textbf{X}_i$ in Eq.2.**
>
> **A2:** For $\textbf{X}\_\{ij\}$ in Eq.5, it denotes the soft weight assigned to each possible human-object pair $ij$ composed by human $i$ and object $j$. The difference between $\textbf{X}\_\{ij\}$ in Eq.5 and $\textbf{X}\_i$ in Eq.2 is that the former encompasses arbitrary human-object interactions composed by given human $i$, action $n$, and object $j$ and reasons over them to compose the final output, while the latter pertains solely to human-object pairs $i$ that have already be established in advance. Evidently, the former delivers predictions in a compositional manner, which facilitates a comprehensive knowledge of different entities and the relations between them.
>
> To improve readability, Eq.2, Eq.5, and Eq.6 are reformulated as follows:
>
> $$
> \textbf{X}'\_{i} =  \textbf{W}^{v'}\cdot\sum\_\{n=1\}^\{N\}\text{softmax}{(\textbf{F}^q\_\{i\}\cdot\textbf{F}^k\_\{n\}/\sqrt{\textit{D}}})\cdot\textbf{F}^v\_\{n\},  \ \ \ \ \ \ \ \ \ \ \text\{(Eq.2)\}
> $$
>
> $$
> \textbf{X}'\_{ij} =  \textbf{W}^{v'}\cdot\sum\_\{n=1\}^\{N_a\}\text{softmax}{(\textbf{F}^q_\{in\}\cdot\textbf{F}^k\_\{nj\}/\sqrt{\textit{D}}})\cdot\textbf{F}^v\_\{inj\},  \ \ \ \ \  \text\{(Eq.5)\}
> $$
>
> $$
> \textbf{Y} = \mathcal{D}^p(\textbf{V}, \textbf{Q}^h, \textbf{Q}^a, \textbf{Q}^o)\in{\mathbb{R}}^{N_h \times N_o \times D},  \ \ \ \  \ \ \ \ \ \ \ \ \ \ \ \ \ \ \ \  \text\{(Eq.6)\}
> $$
>
> where $\textbf{X}$ in Eq.2 and Eq.5 are replaced with $\textbf{X}'$, and $\textbf{X}$ in Eq.6 is replaced with $\textbf{Y}$. $\mathbf{Y}$ is the ultimate HOI prediction produced by the interaction decoder $\mathcal{D}^p$.
>
> ---
>
> **Q3: Typo.**
>
> **A3:** Sincerely thanks for pointing this out! We will correct them in the new version.
>
> ---
>
> **Q4: Describe $\mathcal{G}\_\{v,p\}$ and $\mathcal{G}\_\{o,p\}$ intuitively.**
>
> **A4:** For $\mathcal{G}\_\{v,p\}$, it scores the satisfaction of predictions to rules defined in Eq.7. For example, given a high probability of action "ride" (*i.e.*, a high value of $s\_k[v]$) and the position relationship is "above", if the probability of infeasible interactions (*e.g.*, "human-feed-fish") is high, then the value of $\mathcal{G}\_\{v,p\}$ would be low and $\mathcal{L}\_\{v,p\} = 1 - \mathcal{G}\_\{v,p\}$ should receive a large value so as to punish this prediction. $\mathcal{G}\_\{o,p\}$ is similar but it scores the satisfaction of predictions to Eq.8 with given position and objects such as "horse", "fish", *etc.*
>
> ---
>
> **Q5: Overfitting brought by logic-guided constraints.**
>
> **A5:** **First**, the affordances and proxemics properties (*i.e.*, infeasible interactions with pre-given action, object, or position) are defined according to human knowledge, but not simply retrieved from the annotations of datasets. While the latter tends to overfit the datasets and ignore other possible interactions, the former is all-inclusive.
>
> **Second**, for each rule defined in Eq.7 and Eq.8, it is jointly measured by both action/object and position cues. This can reduce ambiguity. For example, though "human-above-object" can be "person feed a horse from atop it", **L238** also specifies that the possibility of a horse is **LOW** which makes "human-feed-horse" implausible.
>
> **Third**, as stated in **L253**, our logic-guided reasoner learning (LRL) only updates the parameters of the interaction decoder but not the entire network, so as to prevent overfitting.
>
> **Finally**, the zero-shot setup stands as a crucial approach for assessing the generalization capabilities of models [ref1]. As illustrated in Table 4, our logic-guided constraints yield notable performance improvements of **+1.96%**, **+2.41%**, and **+1.43%** across three different zero-shot setups. This demonstrates that the integration of logic-guided constraints not only averts overfitting but also amplifies the generalizability of our method.
>
> Thank you for raising such a good question. We will merge the above discussion into the main text.
>
> [ref1] Multitask Prompted Training Enables Zero-Shot Task Generalization. ICLR22
>
> ---
>
> **Q6: Definition of infeasible interactions.**
>
> **A6:** We construct a large-scale knowledge base wherein the constituents are retrieved from VerbNet [ref1]. To illustrate:
> - First, when presented with a term like "human-launch-XXX", we utilize VerbNet to identify thematic roles associated with the corresponding verb class. These thematic roles signify the semantic relationships between the verb and the arguments. What we are interested in these roles is possible objects or entities that can be "launched".
> - Then, we generate a list of objects based on the thematic roles which can be used as suggestions or options for "xxx" in the context of "human-launch-xxx".
> - Finally, given all of the interactions (*i.e.*, "human-action-object") in HICO-DET or V-COCO, items with action other than "launch" or the objects not found within the aforementioned list are deemed infeasible.
>
> [ref1] VerbNet: A broad-coverage, comprehensive verb lexicon. University of Pennsylvania, 2005.
>
> ---
>
> Thank you for your valuable time and constructive feedback. We will revise the main text according to your review and the discussion above. It is welcome if there are any further questions or suggestions.

---

> > ### Author Response · Authors · 2023-08-20
> > **Looking forward to your further discussion**
> >
> > Dear Reviewer jaUK,
> >
> > Thank you again for your kind review and comments. We hope we have addressed all of your concerns. We wonder if there are any additional comments. We sincerely hope that we will be able to use the remaining time to engage in an open dialogue with domain experts to enhance the quality of our work.
> >
> > Thanks again, Authors

---

### Official Review · Reviewer_AydW · 2023-07-05

**Soundness:** 2 fair
**Presentation:** 1 poor
**Contribution:** 2 fair
**Rating:** 4
**Confidence:** 4

**Summary:**

The paper addresses the problem of human-object interaction (HOI) detection. The authors introduce a new "triplet-reasoning attention" mechanism that enumerates all possible "human-action-object" triplets. Meanwhile, the authors propose a new logic-guided loss function, which enables an interaction detector with stronger reasoning capability. The new method (LoTR) is evaluated on HICO-DET and V-COCO, achieving competitive results.

**Strengths:**

I believe that the authors may introduce some new points to the domain of HOI detection, especifically the logic-guided learning strategy and the proposed LoTR demonstrates excellent performance. However, the paper is really hard to follow due to the poor presentation.

**Weaknesses:**

- In Eq. 4, $\mathbf{Q}^h \in \mathbb{R}^{N_h \times 1 \times D}$ and $\mathbf{Q}^a \in \mathbb{R}^{1 \times N_a  \times D}$, why $\mathbf{Q}^h + \mathbf{Q}^a \in \mathbb{R}^{N_h \times N_a}$
- The $\mathbf{X}$ are *inputs* (line 186) in Eq. 5 and *outputs* (line 191) in Eq. 6.  According to my understanding，the $\mathbf{X}$ in Eq.5 are the outputs of "triplet-reasoning attention" while the $\mathbf{X}$ in Eq.6 are the outputs of the decoder, i.e., the outputs of cross-attention (omitted residual connection and FFN).
- In Eq. 7 and Eq. 8, what is $x(\cdot)$ and what is x? is it a function or a variable? In Eq. 7, what is $M$? And it is really hard for me to understand L214-217. In line 213, $\{ h_1, h_2,\cdots, h_N\}$ are infeasible interactions,  which yet seem to be reasonable interactions in line 217 (e.g., "human-launch-boat").
- Eq. 11 and Eq. 12 need more detailed description. What is $M$ in Eq. 11 and what is $N$ is Eq. 12? Is $h_m$ different from $h_n$? What is $x_k$? The action-position loss, $L_{v,p} = (1-G_{v,p}) \propto {1-s_k(v)[1-s_k(h_m)]}$,
which decreases as $s_k(v)$
increases and $s_k(h_m)$ decreases.  It can be intuitively interpreated that when an action *v* gets a high confidient score, then the interaction $h_m$ should get a low score. How to understand it?



**Questions:**

1. Acorrding to Line 180-182, the "triplet reasoning attention" actually enumerates all human-action-object combinations ($N_h \times N_a \times N_o$), in which most are negative proposals.
2. The number of queries in this paper is $N_h \times N_a \times N_o = 32^3 = 32768$. However, in pervious approaches, the number of queries is normally less than 100. So the computational complexity should be reported.

**Limitations:**

Yes, the authors have addressed the limitations.

---

> ### Author Rebuttal · Authors · 2023-08-09
>
> **Q1: Clarify the dimension of $\textbf{Q}^h$ + $\textbf{Q}^a$.**
>
> **A1:** Sorry for this confusion. We omit a powerful feature named "broadcasting" which is widely used in Pytorch and Numpy.
> Broadcasting allows to perform element-wise operations between matrices of different shapes without explicitly duplicating the data. For example, when adding $\textbf{Q}^h\in\mathbb{R}^{N_h \times 1 \times D}$ to $\textbf{Q}^a\in\mathbb{R}^{1 \times N_a \times D}$, $\textbf{Q}^h$ and $\textbf{Q}^a$ will be firstly duplicated to $\mathbb{R}^{N_h \times N_a \times D}$ along the first and the second dimensions, respectively. The related part will be updated to render a mathematically rigorous statement.
>
> ---
>
> **Q2: Clarify $\textbf{X}$ in Eq.5 and Eq.6.**
>
> **A2:** Apologies for this confusion. To improve readability, we reformulate these two equations as follows:
>
> $$
> \textbf{X}'\_{ij} =  \textbf{W}^{v'}\cdot\sum\_\{n=1\}^\{N_a\}\text{softmax}{(\textbf{F}^q\_\{in\}\cdot\textbf{F}^k_\{nj\}/\sqrt{\textit{D}}})\cdot\textbf{F}^v\_\{inj\}.  \ \ \ \ \  \text\{(Eq.5)\}
> $$
>
> $$
> \textbf{Y} = \mathcal{D}^p(\textbf{V}, \textbf{Q}^h, \textbf{Q}^a, \textbf{Q}^o)\in{\mathbb{R}}^{N_h \times N_o \times D}, \ \ \ \ \ \ \ \ \ \ \ \ \ \ \ \ \ \ \ \ \text\{(Eq.6)\}
> $$
>
> where $\textbf{X}'$ denotes the output of  triplet-reasoning attention and the symbol $\mathbf{Y}$ designates the ultimate output produced by the interaction decoder $\mathcal{D}^p$.
>
> ---
>
> **Q3.1: Clarify $x$ and $x(\cdot)$ in Eq.7 and Eq.8.**
>
> **A3.1:** Sorry for this confusion. As specified in **L214**, the symbol $x$ here denotes a variable representing one human-object pair.  There is no $x(\cdot)$ since Eq.7 and Eq.8 adhere rigorously to the syntax of standard first-order logic formula, where $x$ is closely bound with the universal quantifier $\forall$ to compose $\forall x$, denoting the assertion "for all $x$".
>
> **Q3.2: "Human-launch-boat" seems reasonable in L214-217.**
>
> **A3.2:** Though "human-launch-boat" seems feasible, it is essential to note that the positional cue is confined to "object-above-human" in **L216** (*i.e.*, $p$ is above), thereby rendering the feasibility of "human-launch-boat" unattainable within this context. The related part will be updated to make it more clear.
>
> ---
>
> **Q4.1: Clarify $M$, $N$ and $h$ in Eq.11 and Eq.12.**
>
> **A4.1:** We sincerely apologize for this mistake that the human-object interaction sequence {$h_1, \cdots, h_N$} in **L213** and **L255** should be {$h_1, \cdots, h_M$}. Here $M$ refers to the number of infeasible interactions with pre-given action $v$ and position $p$, while $N$ refers to that with pre-given object $o$ and position $p$.
>
> **Q4.2: Clarify $x_k$ in Eq.11 and Eq.12.**
>
> **A4.2:** $x_k$ represents a human-object pair which is the instantiated value of variable $x$ and $K$ is the number of all human-object pairs.
>
> ---
>
> **Q4.3: Explain $\mathcal{L}_{v,p}$.**
>
> **A4.3:** Note {$h_1, \cdots, h_M$} correspond to the set of **infeasible** interactions with the given verb $v$ (**L213**), which implies a **NEGATIVE** relationship between the verb $v$ and interactions {$h_1, \cdots, h_M$}. In other words, as the score of $v$ increases, the scores of those infeasible interactions {$h_1, \cdots, h_M$} should decrease.
>
> ---
>
> **Q5: Most human-action-object combinations are negative.**
>
> **A5:** We follow the conventions that nearly all existing work [9-20,39-56] utilizes Focal loss for the classification of interactions which is capable of such an imbalanced positive-negative sample ratio. For instance, in HICO-DET, the average count of positive human-object pairs within an image is 4.65 and the possible combination number of our method is $16^3=4096$ (we only keep half of the human, object, and action queries as stated in **L195-196**) for $16^2=256$ human-object pairs (*i.e.*, 4.65 :256). In contrast, the detection task presents a positive-negative ratio of candidate boxes after filtering up to 1:2000 [ref1].
>
> Moreover, our logic-guided reasoner learning (LRL) imposes additional constraints on the combinations of human-object pairs by considering exclusivity, *i.e.*, the interaction cannot be "human-launch-horse" when the action is "launch". This directs the model to focus on positive samples and is substantiated by ablative studies presented in Table 3 that the performance is notably enhanced from **34.32%** to **35.47%**.
>
> [ref1] Focal Loss for Dense Object Detection. ICCV 2017.
>
> ---
>
> **Q6: Larger query number and computation efficiency.**
>
> **A6:** We give a short discussion on the computation efficiency of LoTR in **L356-362**, while a comprehensive comparison to existing work in terms of parameters, FLOPs, and inference speed is elaborated in the *Supplementary Materials* (**i.e.**, Table S1). The summarized results are presented below.
>
> |Method|Params|FLOPs|FPS|HICO-DET|
> |:-|:-|:-|:-|:-|
> |One-stages Detectors:|
> |PPDM|194.9|-|17.14|21.73|
> |HOTR|51.2|90.78|15.18|23.46|
> |QPIC|41.9|88.87|16.79|29.07|
> |CDN-S|42.1|-|15.54|31.78|
> |GEN-VLKs|42.8|86.74|18.69|33.75|
> |LoTR|49.8|89.65|16.84|**35.47**|
>
> It can be seen that the params, FLOPs, and inference speed of our method align closely with existing one-stage work. The rationale behind this is discussed in **L356-362**: **i)** as stated in **L195-196**, we only keep **half** the number of human, object, and action queries by filtering the low-scoring ones before sending them into the interaction decoder, resulting maximum $16^3=4096$ human-action-object triplets, **ii)**, the layer number of Transformer decoder in LoTR is set to 3, which is the **smallest** among existing work, and **iii)**  triplet-reasoning attention introduces nearly no additional parameters.
>
> ---
>
> Thank you for providing such a comprehensive and detailed review. We are committed to reformulating our notation system and revising the methodology section in accordance with your valuable suggestions. Please do not hesitate to post comments if there are any further questions we can help.

---

> > ### Author Response · Authors · 2023-08-20
> > **Looking forward to your further discussion**
> >
> > Dear Reviewer AydW,
> >
> > Thank you again for your kind review and comments. We hope we have addressed all of your concerns. We wonder if there are any additional comments. We sincerely hope that we will be able to use the remaining time to engage in an open dialogue with domain experts to enhance the quality of our work.
> >
> > Thanks again, Authors

---

> > ### Comment · Area_Chair_Cwny · 2023-08-20
> > **Please respond**
> >
> > Dear AydW,
> >
> > Given you are in the minority with a reject it is critical that you read the rebuttal and inquire about any outstanding issues you feel are still not addressed.
> >
> > Thank you,
> > AC

---

### Official Review · Reviewer_mDiH · 2023-07-07

**Soundness:** 3 good
**Presentation:** 3 good
**Contribution:** 3 good
**Rating:** 5
**Confidence:** 4

**Summary:**

The paper presents LoTR, a method for detecting human-object interactions (HOI) that utilizes Transformer as a reasoning mechanism to model the interactions between people, objects and their interactions. By feeding the self-attention with different feature combination as key query and value, LoTR can infer triplets and establishes meaningful interactions. The paper demonstrates the effectiveness of LoTR through experimental results on two widely used datasets, showcasing its strong performance.

**Strengths:**

- The paper is well written and well motivated.
- The idea of Triplet-Reasoning Attention is well motivated and seems working as expected.
- The results are comprehensive (I do have concerns on the results as well, please see section below).

**Weaknesses:**

1. Some of the claims may lack sufficient support or appear too strong. For instance, in lines 30-33, it is advisable to provide citations or evidence to substantiate the claim. It seems contradictory to assert that the model can accurately identify the correct pair while lacking comprehension of them.

2. The comparison of results in Table 1 and Table 2 lacks fairness. As indicated in line 249, this paper utilizes the CLIP model, whereas most of the baselines are trained on significantly smaller datasets. The authors should establish a fair comparison by employing the exact same backbone or explicitly state the pre-training set used for the backbones.

3. Additional discussion and analysis are necessary. For instance, when the proposed model performs worse than STIP on V-COCO, what could be the potential reasons? Currently, the paper criticizes STIP for its performance on other datasets but fails to explain why the proposed method does not generalize as well to V-COCO.

4. In experimental results section, more details are required. For example, what does the bolded text signify in Table 5? Additionally, when the authors claim that they selected a three-layers decoder for improved efficiency, what are the corresponding FLOPs or latencies associated with different configurations?

**Questions:**

Please see my comments above.

---

> ### Author Rebuttal · Authors · 2023-08-09
>
> **Q1: Claims may be too strong and lack sufficient support.**
>
> **A1:** Thank you for pointing this out! The related parts will be revised as follows:
>
> "**First**, the principal focus of them is to optimize the samples with known concepts, ignoring a large number of feasible combinations that were never encountered within the training dataset, resulting in poor zero-shot generalization ability [ref1]. **second**, the human-object pairs are usually proposed by a simple MLP layer[21,39-53] or simultaneously constructed when predicting humans and objects[10-17], without explicit modeling of the complex relationships among subjects, objects, and the ongoing interactions that happened between them."
>
> For the first limitation, we supplement the training objectives with logic-guide constraints. These constraints are converted from the affordances and proxemics properties, effectively furnishing the model with ongoing guidance, even for instances never seen before.
>
> For the second limitation, we decouple the inputs of interaction decoders into three different kinds of entities (*i.e.*, "human", "object" and "action"), and the final predictions (*i.e.*, "human-action-object") are then obtained through the application of triplet-reasoning attention, which takes these three items as input and facilitates reasoning over them.
>
> [ref1] Discovering Human-Object Interaction Concepts via Self-Compositional Learning. ECCV 2022.
>
> ---
>
> **Q2: Comparison to work without CLIP.**
>
> **A2:** Good suggestion! To enable a fair comparison, we additionally experiment with our proposed method without using visual-linguistic pre-trained models like CLIP. The results under the zero-shot setup are summarized below. It emerges that even in the absence of CLIP, LoTR can still produce top-leading performance. Please note that FCL[98], ATL[99], and SCL[100] are **meticulously tailored** for zero-shot HOI detection, entailing much more intricate training strategies such as fabricated pair generation, pseudo labeling, and self-supervised learning, *etc.*
>
> |Method|Type|VL Pretrain|Unseen|Seen|Full|
> |:-|:-|:-|:-|:-|:-|
> |FCL[98]|RF-UC|-|13.16|24.23|22.01|
> |SCL[100]|RF-UC|-|19.07|30.39|28.08|
> |**LoTR**|RF-UC|-|**21.95**|**32.81**|**30.62**|
> |GEN-VLKT[17]|RF-UC|CLIP|21.36|32.91|30.56|
> |**LoTR**|RF-UC|CLIP|**25.97**|**34.93**|**33.17**|
> |FCL[98]|NF-UC|-|18.66|19.55|19.37|
> |SCL[100]|NF-UC|-|21.73|25.00|24.34|
> |**LoTR**|NF-UC|-|**23.68**|**25.23**|**24.87**|
> |GEN-VLKT[17]|NF-UC|CLIP|25.05|23.38|23.71|
> |**LoTR**|NF-UC|CLIP|**26.84**|**27.95**|**27.86**|
> |ATL[99]|UO|-|5.05|14.69|13.08|
> |FCL[98]|UO|-|0.00|13.71|11.43|
> |**LoTR**|UO|-|**8.41**|**21.35**|**18.48**|
> |GEN-VLKT[17]|UO|CLIP|10.51|28.92|25.63|
> |**LoTR**|UO|CLIP|**15.67**|**30.42**|**28.23**|
> |GEN-VLKT[17]|UV|CLIP|20.96|30.23|28.74|
> |**LoTR**|UV|CLIP|**24.57**|**31.88**|**30.77**|
>
> We also give experiments under the regular HOI detection setup without using CLIP. Evidently, our LoTR continues to exhibit remarkable performance, up to $1.02$ mAP improvement over IF-HOI[17] which does not utilize CLIP.
>
> |Method|Backbone|VL Pretrain|Full|Rare|Non-Rare|
> |:-|:-|:-|:-|:-|:-|
> |UPT[15]|R50|-|31.66|25.94|33.36|
> |STIP[21]|R50|-|32.22|28.15|33.43|
> |ODM[106]|R50-FPN|-|31.65|24.95|33.65|
> |Iwin[107]|R50-FPN|-|32.03|27.62|34.14|
> |IF-HOI[17]|R50|-|33.51|30.30|34.46|
> |**LoTR**|R50|-|**34.53**|**31.12**|**35.38**|
> |SSRT[60]|R50|CLIP|30.36|25.42|31.83|
> |CATN[63]|R50|CLIP|31.71|24.82|33.77|
> |DOQ[62]|R50|CLIP|33.28|29.19|34.50|
> |GEN-VLKT[17]|R50|CLIP|33.75|29.25|35.10|
> |**LoTR**|R50|CLIP|**35.47**|**32.03**|**36.22**|
>
> These experiments solidly demonstrate the effectiveness of our method under both zero-shot and regular setups, even compared with the meticulously tailored counterparts.
>
> ---
>
> **Q3: Inferior performance on V-COCO compared to STIP[21].**
>
> **A3:** **First**, this may be attributed to the inherent characteristics of the V-COCO dataset, comprising merely 29 action categories, 80 object categories, and 263 kinds of human-object interactions. In light of this constrained diversity, the efficacy of our introduced logic-guided learning, which incorporates a large amount of affordances and proxemics properties into model reasoning, becomes especially modest.
>
> **Second**, STIP is a two-stage method that relies on a simple MLP to propose positive human-object pairs. Such a proposal network may achieve promising performance when the number of interaction classes is limited and exhibits inefficiency when applied to datasets with extensive interaction categories. Therefore, we experiment with the accuracy of the proposal network (percentage of successfully proposed positive human-object pairs) in STIP with the different number of interaction categories on HICO-DET:
>
> |#Category of Interaction|Accuracy|
> |:-:|:-|
> |20%|0.86|
> |40%|0.81 (-0.05)|
> |60%|0.73 (-0.08)|
> |80%|0.60 (-0.13)|
> |100%|0.45 (-0.15)|
>
> Here 20% means the training set only contains 0.2$\times$600 classes of interactions in HICO-DET. As seen, the accuracy of the proposal network experiences an accelerated decline (*i.e.*, performance drops at a larger pace from 20% to 100%) when the number of interaction categories increases, validating our hypothesis above.
>
> ---
>
> **Q4: Detailed elaboration to Table 5.**
>
> **A4:** In Table 5, we would like to probe the impact of different numbers of decoder layers and queries on the performance of LoTR. The **bolded** rows denote the setups that we finally adopted. As suggested, we add the statistics of params, FLOPs, and FPS for different configurations:
>
> |Layer(L)|Params|FLOPs|FPS|Full|Rare|Non-Rare|
> |:-|:-|:-|:-|:-|:-|:-|
> |2|45.4|82.11|18.62|34.61|30.72|35.54|
> |**3**|**49.8**|**89.65**|**16.84**|**35.47**|**32.03**|**36.22**|
> |4|54.3|97.27|14.75|35.37|31.96|36.09|
> |6|63.1|112.57|11.46|35.61|32.13|36.39|
>
> ---
>
> Thank you so much for your careful review. We will revise our manuscript according to your suggestive comments and please feel free to post your feedback if you have any further questions.

---

> > ### Comment · Reviewer_mDiH · 2023-08-16
> > **Thanks for the rebuttal**
> >
> > The rebuttal addressed most of my concerns, please include the additional information into the final paper. I will raise my rating.

---

> > > ### Author Response · Authors · 2023-08-16
> > > **Thanks for the response**
> > >
> > > Thank you for your thoughtful review and for taking the time to consider our rebuttal. We are delighted to hear that the additional information addressed your concerns, prompting a rise in the rating. Your feedback is invaluable in improving the quality of our paper and we are committed to incorporating your suggestions for a new version that reflects the changes.

---

### Author Rebuttal · Authors · 2023-08-09

# To all reviewers

We express our sincere gratitude to all reviewers for their valuable time and thorough assessment of our manuscript. In response, we have meticulously addressed each concern raised, providing point-to-point clarifications which shall be integrated into the new version of our manuscript.

We are gratified by the unanimous recognition from all reviewers that our logic-guided reasoning approach is interesting and novel, bringing significance within the context of the HOI detection task.

Foremost among the feedback is the concern about the writing of our manuscript, particularly with regard to the claim and elaboration of methodology. We acknowledge this and pledge to render a comprehensive revision of our article which will encompass rectifying overclaims, substantiating our conclusions through referencing literature, reformulating the symbol system for enhanced clarity, and providing a more detailed interpretation of the equations presented.

Additionally, in accordance with the above, Reviewer mDiH and gPtK also raise concerns about the evaluation of our method. We appreciate this and have refined both Table 1 and Table 2 to ensure a fair comparison with existing works. The analysis regarding params, FLOPS, and inference speed provided in *Supplementary Material* will be merged into Table 2, thereby affording an all-inclusive evaluation of our method. Moreover, we have given an analysis to the improvement in zero-shot setups brought by each component and tested the generability of our method by incorporating it into two existing methods.

Reviewer uwRp asks us about the application of logic-guided reasoning framework to other computer vision tasks. We are pleased to probe this as it takes the opportunity to prove the potential positive impact of our method to the general computer vision community. The results are also exciting, *i.e.*, substantial enhancements over the baseline performance in two tasks: Visual Question Answering (VQA) and weakly supervised semantic segmentation.

---

### Decision · Program_Chairs · 2023-09-21

**Decision:**

Accept (poster)

**Comment:**

The AC and reviewers appreciate the authors rebuttal and discussion. All reviews, rebuttals and comments were carefully read and considered by the AC. The AC met with the senior AC to carefully review this paper. The reviewers agreed that the paper presents a new and interesting perspective for the HOI problem. For example gPtK wrote: "This paper takes a further step by converting the rule-based logic into differentiable operations, which is very interesting."  Reviewers found the paper mostly well written and the experimental section detailed and informative. Due to this the AC recommends that this paper be accepted to NeurIPS.

However a number of the reviewers found some equations and details about the method to be confusing. Through the rebuttal process most of these issues have been clarified but the AC asks that the authors be sure to include these fixes in the final camera ready version of the paper to insure it maintains the NeurIPs bar for quality and correctness.